# Emerging negative impact of warming on summer carbon uptake in northern ecosystems

Tao Wang[1,2], Dan Liu[2], Shilong Piao [1,2,3], Yilong Wang[4], Xiaoyi Wang[2], Hui Guo[2], Xu Lian[3], John F Burkhart [5], Philippe Ciais [4], Mengtian Huang[3], Ivan Janssens[6], Yue Li [3], Yongwen Liu [3], Josep Peñuelas [7,8], Shushi Peng [3], Hui Yang[3], Yitong Yao[3], Yi Yin[4] & Yutong Zhao[2]

Most studies of the northern hemisphere carbon cycle based on atmospheric $CO_2$ concentration have focused on spring and autumn, but the climate change impact on summer carbon cycle remains unclear. Here we used atmospheric $CO_2$ record from Point Barrow (Alaska) to show that summer $CO_2$ drawdown between July and August, a proxy of summer carbon uptake, is significantly negatively correlated with terrestrial temperature north of 50° N interannually during 1979–2012. However, a refined analysis at the decadal scale reveals strong differences between the earlier (1979–1995) and later (1996–2012) periods, with the significant negative correlation only in the later period. This emerging negative temperature response is due to the disappearance of the positive temperature response of summer vegetation activities that prevailed in the earlier period. Our finding, together with the reported weakening temperature control on spring carbon uptake, suggests a diminished positive effect of warming on high-latitude carbon uptake.

[1] Center for Excellence in Tibetan Earth Science, Chinese Academy of Sciences, Beijing 100085, China. [2] Key Laboratory of Alpine Ecology, Institute of Tibetan Plateau Research, Chinese Academy of Sciences, Beijing 100085, China. [3] Sino-French Institute for Earth System Science, College of Urban and Environmental Sciences, Peking University, Beijing 100871, China. [4] Laboratoire des Sciences du Climat et de l'Environnement, LSCE, Gif-sur-Yvette F-91191, France. [5] Department of Geosciences, University of Oslo, Oslo 0371, Norway. [6] Department of Biology, University of Antwerp, Universiteitsplein 1, 2610 Wilrijk, Belgium. [7] CREAF, Cerdanyola del Valles, Barcelona 08193 Catalonia, Spain. [8] CSIC, Global Ecology Unit CREAF-CEAB-CSIC-UAB, Cerdanyola del Valles, Barcelona 08193 Catalonia, Spain. Correspondence and requests for materials should be addressed to T.W. (email: twang@itpcas.ac.cn)

Arctic and boreal ecosystems play an important role in the global carbon cycle, and their carbon cycle responses to climate change become a major global concern[1,2]. The relationship between the carbon cycle and temperature in these ecosystems has received increasing attention, because temperature plays a dominant role in controlling net primary productivity (NPP)[3], respiration[4,5], and fire disturbances[6] at these high latitudes. Numerous field studies in high-latitude ecosystems have been conducted in the summer[7–15], but the results of these local experiments are limited in extent and are difficult to scale-up to larger regions. Observations of atmospheric $CO_2$ concentration from atmospheric stations at high latitudes provide complementary monitoring of the dynamics of carbon exchange in northern ecosystems. But most of these studies have concentrated on the amplitude of the seasonal cycle or the spring and autumn boundaries of the growing season[16–18], little attention has been given to the summer. Summer has the highest vegetation productivity, strongly contributes to interannual variations in terrestrial carbon uptake[19] and generally has the most favorable climatic conditions for $CO_2$ uptake.

Observations suggest that the rates of climate change on a decadal scale have varied in the last three decades[20], with still imperfectly known implications for changes in carbon cycling[17,21,22]. Mounting evidence suggests that the response of the carbon cycle to temperature at high northern latitudes is not constant over time[23,24], but this evidence is restricted to studies of vegetation productivity proxies[21,22] and spring carbon fluxes[17]. Much less is known about summer $CO_2$ uptake. Understanding temporal changes in the relationship between summer $CO_2$ uptake (an indicator of maximum $CO_2$ uptake capacity) and temperature is important for determining the contribution of future northern terrestrial carbon fluxes to accelerating or decelerating ongoing warming.

The aim of this study is to understand the effect of temperature on summer carbon uptake in northern ecosystems and its decadal variation. We used the long-term record of atmospheric $CO_2$ concentrations from the Barrow atmospheric $CO_2$ monitoring station (71°N, 157°W, Alaska)[25] to calculate the summer $CO_2$ drawdown (SCD), which was used as an indicator of summer carbon uptake. SCD was calculated as the difference in the $CO_2$ concentration between the first week of July and the last week of August in the detrended $CO_2$ record (see Methods, Supplementary Fig. 1). Here, with simultaneous use of multiple satellite-derived products[26–28], an ensemble of terrestrial carbon cycle models and simulations with an atmospheric transport model[29] (Methods), we show that there is a significant-negative interannual correlation between SCD with summer land temperature north of 50°N during 1979–2012, and such significant negative correlation only occurred during the earlier period (1979–1995) instead of during the later period (1996–2012). This emergent-negative relationship is primarily due to summer vegetation activities no longer positively responding to temperature.

## Results

**Emerging-negative temperature control on summer $CO_2$ drawdown.** An analysis of the interannual correlation between SCD and terrestrial summer temperature north of 50°N ($R_{SCD-T}$), whereas controlling for the effects of other summer climatic variables such as precipitation and cloudiness (all variables detrended, see Methods), indicated a negative correlation of SCD with temperature ($R = -0.46$, $P < 0.01$) for the full length of the study period, 1979–2012. In other words, warmer years were associated with a lower carbon uptake. The sensitivity of SCD to the interannual variability of summer temperature ($\gamma_{SCD-T}$) was calculated as the slope of the regression line in a multiple regression of SCD against temperature, precipitation, and cloudiness in summer for 1979–2012 (Methods). The results of the regression tell us that

an increase of 1 °C in summer temperature leads to a reduced SCD of 1.45 ppm $CO_2$ at Barrow.

We further analyzed the decadal changes in $R_{SCD-T}$ and found strong differences between the earlier (1979–1995) and later (1996–2012) periods of the Barrow time series (Fig. 1a). $R_{SCD-T}$ was small and not statistically significant ($R = -0.40$, $P = 0.14$) in the earlier period, but became statistically significant and more negative ($R = -0.65$, $P < 0.01$) in the later period. The relationship between SCD and temperature during the later period was thus responsible for the negative impact of temperature on summer $CO_2$ drawdown during the whole record. We also found that $\gamma_{SCD-T}$ became more negative, decreasing from $-1.23 \pm 0.76$ ppm year$^{-1}$ °C$^{-1}$ for 1979–1995 to $-2.06 \pm 0.46$ ppm year$^{-1}$ °C$^{-1}$ for 1996–2012 (Supplementary Fig. 2). The amplitude of the seasonal $CO_2$ concentration at Barrow was found to lag behind temperature by about 2 years[30], and it was suggested that this lag was due to a lag in the response of net primary production to temperature. In contrast, SCD was not significantly correlated with summer temperature in the previous 2 years (or 1 year) for any of the study periods (1979–2012, or the two periods 1979–1995 and 1996–2012). This lack of lagged-correlation coincides with a non-significant lagged-response of summer productivity to temperature in the previous one or 2 years (Supplementary Fig. 3). This result does not contradict the result from Keeling et al.[30], but it shows that if there is a lagged response of the peak-to-peak $CO_2$ amplitude to temperature, it is not due to a lag of summer $CO_2$ uptake.

**Robustness tests.** We carried out a variety of tests to check the robustness of the shift in $R_{SCD-T}$ and its sensitivity to changes in the data and methods used. We first calculated $R_{SCD-T}$ using climatic variables spatially weighted with the footprint intensity over the flux footprint area of the $CO_2$ record of the Barrow station (Methods), because terrestrial carbon fluxes impacting the Barrow $CO_2$ concentration are not spatially uniform in the high-latitudes. The area of the summer flux footprint of the Barrow station calculated using the FLEXPART Lagrangian particle dispersion model[31] was mainly restricted to the regions of Siberia and Alaska and the Chukchi and Beaufort Seas (Supplementary Fig. 4). A similar footprint area was found by a simulation of the sensitivity of the Barrow $CO_2$ measurements during the last week of August to terrestrial carbon fluxes 20 days before the measurements using the adjoint code of the Laboratoire de Météorologie Dynamique (LMDZ) atmospheric transport model (see Methods; Supplementary Fig. 5a–c). We also performed 40- and 60-day back-trajectory calculations using LMDZ and found that the footprint area for fluxes influencing the SCD did not change significantly compared to a 20 days influence (Supplementary Fig. 5). $R_{SCD-T}$ between the two study periods still decreased when calculated using temperature spatially weighted with the footprint intensity. $R_{SCD-T}$ shifted from a non-significant-positive value ($R = 0.01$, $P = 0.97$) to a significant-negative value ($R = -0.61$, $P < 0.05$) in this test (Fig. 1b). $R_{SCD-T}$ also shifted when the footprint was derived from 40- to 60-day back-trajectory calculations rather than with a value of 20 days (Supplementary Fig. 6).

To test whether the observed shift in $R_{SCD-T}$ is real or an artifact caused by extreme years, we analyzed the frequency distribution of $R_{SCD-T}$ obtained by randomly selecting 14 years for each period and found that $R_{SCD-T}$ decreased from $-0.42 \pm 0.09$ during 1979–1995 to $-0.64 \pm 0.10$ during 1996–2012 (Fig. 1a), suggesting that the presence of extreme years was not responsible for the shift in $R_{SCD-T}$. $R_{SCD-T}$ also shifted when we used a different climatic dataset (Supplementary Fig. 7) and weekly rather than daily Barrow $CO_2$ concentration data (Supplementary Fig. 8). $R_{SCD-T}$ again decreased significantly for 1979–2012 ($P < 0.01$) when calculated with a 15-year moving window (with both SCD and temperature detrended) (Supplementary Fig. 9).

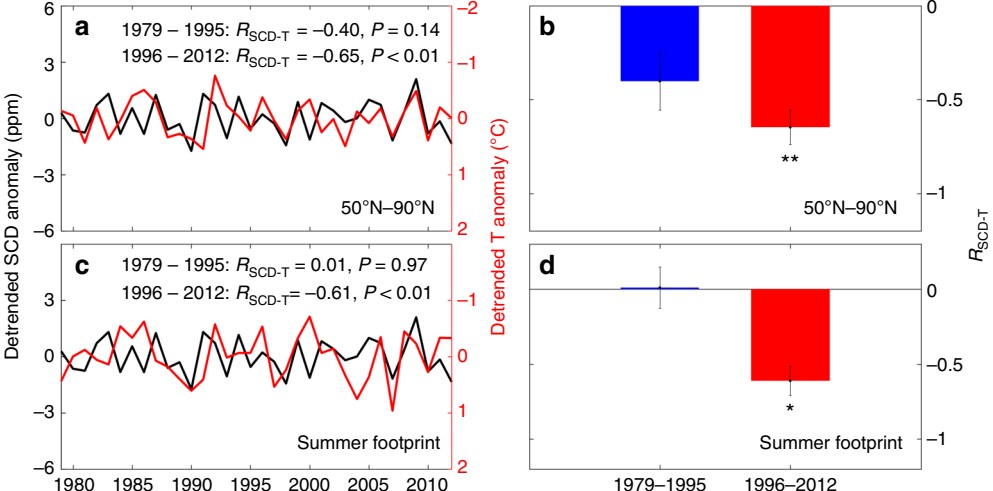

**Fig. 1** Negative temperature control of summer $CO_2$ drawdown. **a–d** Time series of anomalies of summer $CO_2$ drawdown (SCD, black line) and summer temperature (T, red line) calculated as the average for July and August across ecosystems north of 50°N (**a**), the spatial average weighted by the potential emission sensitivities from FLEXPART over the vegetated land area within the multi-year mean summer footprint (**c**), The comparison of interannual partial-correlation coefficient between SCD and T ($R_{SCD-T}$) between the two periods (1979–1995 and 1996–2012) based on north of 50°N and summer footprint, respectively **b** and **d**. The interannual partial-correlation coefficient is calculated by statistically controlling for the effects of summer precipitation and cloudiness. We calculate $R_{SCD-T}$ through randomly selecting 14 of the 17 years in each corresponding period, and then take their standard deviation as the error bar. All variables were detrended for each period before the partial-correlation analysis. * and ** indicate that the partial-correlation coefficient is significant at $P < 0.05$ and $P < 0.01$, respectively. The figure was created using Matlab R2016a

We also explored the robustness of the result to the method used to calculate SCD. SCD was initially calculated as the difference in the $CO_2$ concentration between the first week of July and the last week of August in the detrended $CO_2$ record, but $CO_2$ uptake decreased briefly in August (Supplementary Fig. 1), which could conceivably impact the results. Therefore, we calculated SCD using alternative methods. We first set SCD as the difference in $CO_2$ concentration between the climatological day of the year when the detrended $CO_2$ concentration crossed its long-term mean downwards (climatological spring zero-crossing date) and the climatological day of the year when detrended $CO_2$ concentration reached its annual minimum (climatological trough date). $R_{SCD-T}$ calculated from this different definition of SCD varied from $-0.28$ ($P = 0.32$) during 1979–1995 to $-0.67$ ($P < 0.01$) during 1996–2012 (Supplementary Fig. 10), similar to the original SCD definition. The spring zero-crossing date and the trough date for 1979–2012 both advanced at rates of 0.22 and 0.21 day year$^{-1}$, respectively (Supplementary Fig. 11), so a fixed window for defining SCD using the climatological zero-crossing and trough dates may not be appropriate. We thus performed an additional analysis that allowed the spring zero-crossing and trough dates to vary interannually. In this additional analysis, we found that $R_{SCD-T}$ shifted from 0.42 ($P = 0.12$) during 1979–1995 to $-0.52$ ($P < 0.01$) during 1996–2012 (Supplementary Fig. 12). During the earlier period, $R_{SCD-T}$ becomes positive instead of staying negative as it does when using the interannually varying window to define SCD. But the significant-negative correlation between SCD and temperature still emerged in the later period leading to the main conclusion that this result is not affected by the method used to define SCD.

We further investigated whether the shift in $R_{SCD-T}$ may have been indirectly related to changes in the onset date of the Arctic sea-ice melt. The date of the onset of melting of sea ice advanced at rates around 2.8 day decade$^{-1}$ during 1979–2004 at Chukchi and Beaufort Seas[32]. Earlier melting could increase the air-sea $CO_2$ flux in the summer and potentially contribute to the variations of summer $CO_2$ concentration at Barrow. A possible

effect of Arctic sea-ice melt on the shift in $R_{SCD-T}$ was then investigated by using the partial-correlation between SCD and land temperature after controlling for the effects of cloudiness, precipitation, and summer Arctic sea-ice extent (SIE). $R_{SCD-T}$ shifted similarly to the original calculation (Supplementary Fig. 13), suggesting that the impact of earlier Arctic sea-ice melt on the shift in $R_{SCD-T}$ was limited.

**Mechanisms.** We propose two hypotheses for the shift in $R_{SCD-T}$ between the two study periods. Multiple lines of evidence indicate that plant productivity is a major driver affecting interannual variations in net carbon uptake in Arctic and boreal ecosystems[30,33,34], so our first hypothesis is that summer vegetation activities has become less positively responsive to temperature. Our second hypothesis is that the increased response of respiration to temperature, which is typically positive in Arctic and boreal ecosystems, produced the change in $R_{SCD-T}$.

We tested the first hypothesis by calculating the correlation coefficient ($R_{NDVI-T}$) of the interannual relationship between the summer Normalized Difference Vegetation Index (NDVI) (as a proxy for productivity) and terrestrial summer temperature for the two periods (see Methods). We found that NDVI and temperature across ecosystems north of 50°N were always significantly positively correlated during the earlier period ($R = 0.82 \pm 0.07$), but not the later period ($R = 0.08 \pm 0.19$) (Fig. 2a), supporting this hypothesis. A further analysis of the spatial pattern of changes in $R_{NDVI-T}$ indicated that the recent decrease in $R_{NDVI-T}$ occurred over a wide region comprising eastern Siberia and Alaska (Fig. 2d), which constitutes most of the summer footprint of the Barrow station (Supplementary Fig. 4). In addition, analysis of an extended solar-induced chlorophyll fluorescence (SIF) dataset[35] also showed that the partial correlation between SIF and temperature across ecosystems north of 50°N was not significant during the period 2001–2012 ($R_{SIF-T} = -0.19$, $P = 0.61$) (Supplementary Fig. 14). A 30-year global dataset of a satellite-derived NPP model[27], also based on NDVI (Fig. 2b, e) and of gross primary productivity (GPP) observation-based results for

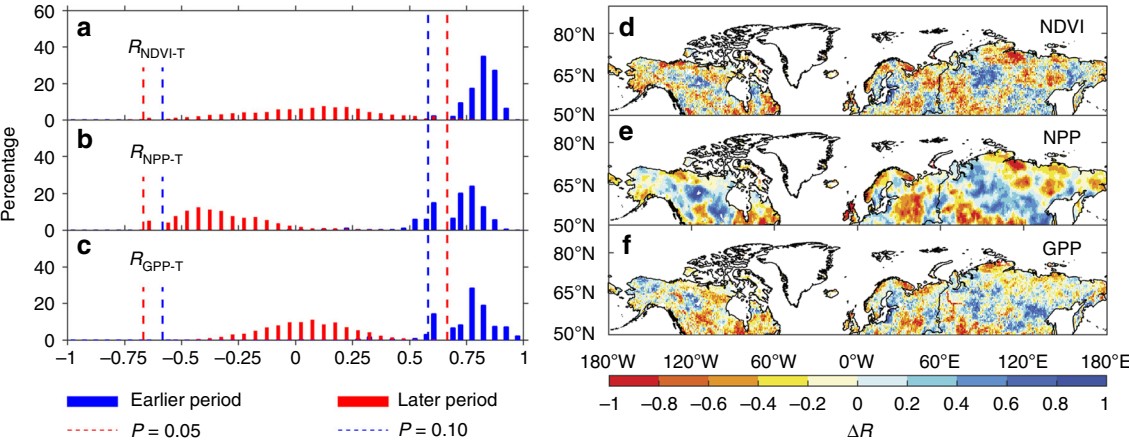

**Fig. 2** Link between summer vegetation activities and summer temperature. **a–c** Frequency distributions of the partial-correlation coefficient of summer NDVI ($R_{NDVI-T}$) (**a**), satellite-derived net primary productivity ($R_{NPP-T}$) (**b**) and gross primary productivity based on flux-tower data ($R_{GPP-T}$) (**c**) with summer temperature (T) during the earlier and later periods. Significant partial-correlation coefficients (based on a sample size of 11) are identified as dashed lines (magenta, $P < 0.05$; blue, $P < 0.1$). All variables were detrended for each period before the partial-correlation analysis. **d–f** Spatial distribution of differences of $R_{NDVI-T}$, $R_{NPP-T}$, and $R_{GPP-T}$ between the earlier and later periods. The earlier period is 1982–1995, and the later periods are 1996–2012 for NDVI, 1996–2011 for NPP and 1996–2011 for GPP. The figure was created using Matlab R2016a

1982–2011 based on flux-tower data[28] (Fig. 2c, f) confirmed the loss of a positive correlation with summer temperature. By contrast, only two of nine terrestrial ecosystem models properly captured the loss of the positive correlation between NPP and temperature (Supplementary Fig. 15).

The decrease in $R_{NDVI-T}$ may have been due to an increase in the number of extreme warm days, because vegetation growth is reduced by heat stress during extremely warm days. The number of extreme warm days (defined as temperatures higher than the 90th percentile of the July and August temperature distribution for 1982–2012) increased during the later period (Supplementary Fig. 16a). The patterns of the changes of $R_{NDVI-T}$ were roughly consistent with those of the number of extreme warm days, particularly Alaska and eastern Siberia constituting the main footprint area of summer $CO_2$ changes at Barrow (Supplementary Fig. 4). NDVI in these areas had a significant-negative partial correlation with the number of extreme warm days when the data were statistically controlled for the effect of mean summer temperature (Supplementary Fig. 16b). The results are robust to the use of sub-daily temperature records in the definition of extreme warm temperatures (Supplementary Fig. 17). A lower $R_{NDVI-T}$ could also be due to a nonlinear response of photosynthesis to temperature, with summer temperatures exceeding the optimum and thereby decreasing photosynthesis[36] and thereby weakening the temperature-productivity correlation when warmer years become more common[21].

The decrease in $R_{NDVI-T}$ may also have been caused by lower soil-moisture contents during the later study period. We investigated this possibility using the values of root-zone soil moisture from the observation-based model of land water budgets constrained by multiple satellite datasets, including surface soil moisture detected by microwave sensors[37,38]. These values of soil moisture in summer were slightly higher during the later than the earlier period (Supplementary Fig. 18), suggesting that changes in soil moisture alone did not account for the decrease in $R_{NDVI-T}$. In addition, increased temperature could induce plant water stress, through increasing atmospheric deficits, to such a degree that plants either lose water at a faster rate or close stomata. We also analyzed changes in the atmospheric vapor pressure deficit (VPD), which is indicative of atmospheric demand for water, between the earlier and later period. Our results showed that

atmospheric demand for water generally increased over the main footprint area of summer $CO_2$ changes at Barrow (Supplementary Fig. 19a). But NDVI in these areas had either a positive or non-significant-negative partial correlation with VPD changes when the data were statistically controlled for the effect of mean summer temperature (Supplementary Fig. 19b). This result suggests that increased atmospheric demand for water has a relatively low probability of imposing water stress constraints on plant growth over high-latitude ecosystems. Therefore, change in VPD should not be the main cause of the observed decrease in $R_{NDVI-T}$.

In contrast to changes in vegetation productivity being indirectly monitored with remote sensing proxies, no satellite observations were available for use directly, or as proxies, to quantify changes in the response of terrestrial respiration to temperature. To test the hypothesis that the loss of temperature dependence of terrestrial respiration was responsible for the change in $R_{SCD-T}$, we used a global respiration dataset that integrates a global soil respiration database[5] with a climate-driven empirical model of soil respiration[39]. The interannual partial correlation of summer heterotrophic respiration (HR) with summer temperature ($R_{HR-T}$), while controlling for the effects of precipitation and cloudiness, was significant for both study periods. The coefficients are $0.85 \pm 0.11$ for 1979–1995 ($P < 0.01$) and $0.57 \pm 0.16$ for 1996–2012 ($P < 0.05$), respectively (Supplementary Fig. 20). Our analysis suggested that respiration continued to respond significantly positively to temperature in both study periods, and the response was not stronger during the later period, implying that the change in $R_{SCD-T}$ was not due to changes in the response of terrestrial respiration to temperature.

## Discussion

We demonstrated that the effect of temperature on summer $CO_2$ drawdown has changed in the last 3 decades, implying a shift in the response of summer carbon uptake to warming in Arctic and boreal ecosystems. A significant-negative effect of temperature on summer $CO_2$ drawdown has recently emerged, that is, warmer years coincide with less summer uptake, which is most likely due to the reduced effect of temperature on summer vegetation activities. This evidence prompts us to re-examine the long-standing paradigm that vegetation activities in high-latitude

ecosystems is limited by temperature and that warming is therefore conducive to increased net carbon uptake[10–12,14,15,40,41]. An earlier study reported that the strong positive correlation between temperature and spring $CO_2$ uptake[30] disappeared during 1996–2012[17], due to a lower response of spring NPP to temperature. The effect of temperature on $CO_2$ drawdown during the summer carbon uptake period thus became negative in the later period. If extrapolated to future warming in the next decades, it leads us to hypothesize that further warming could fundamentally alter high-latitude terrestrial carbon balances, reducing ecosystems' capacity to sequester atmospheric $CO_2$ and ultimately accelerating climate change. In addition, the overall decline in the stimulating effect of temperature on carbon uptake in spring and summer suggested that the previously reported warming-induced increase in the duration of carbon uptake and then the seasonal $CO_2$ amplitude[30] would become weak, consistent with a recent study demonstrating that the increase in seasonal $CO_2$ amplitude was due much more to $CO_2$ fertilization than to climate change[17].

Our finding will add further complexity to the estimation of the carbon balance over high-latitude ecosystems in a warming world. The large temperature-induced shift in carbon uptake during the main period of carbon uptake would reduce the positive effects of the gradual increasing atmospheric concentration of $CO_2$[42], and increased nitrogen deposition[43] on net $CO_2$ uptake. Warming in these northern ecosystems, however, will continue and lead to the thawing of permafrost, triggering a substantial loss of permafrost carbon[44], further aggravating the negative impact of temperature on net $CO_2$ uptake during the main period of carbon uptake. Furthermore, most models of terrestrial ecosystems do not correctly identify the response of productivity to temperature variability over the last three decades, probably due to inaccurate model representations of the impact of extreme hot days and the nonlinear response of photosynthesis to temperature. Accurate model predictions of future feedbacks between the high-latitude carbon fluxes and the climate will require better parameterizations of the processes driving the response of Arctic and boreal ecosystems to warming.

We should stress that understanding of the emerging negative temperature control on summer carbon uptake and its mechanisms is still limited. It would be more straightforward to use net surface $CO_2$ flux data, instead of atmospheric $CO_2$ concentration data, in depicting the relationship between the carbon cycle and climate. But the availability of reliable long-term flux data is currently extremely limited by the sparsity of the in situ observing effort over arctic and boreal regions. Continued efforts are required to increase in situ carbon-cycle observations over these high-latitude ecosystems, to develop more mechanistic ecosystem models and to improve inverse models of assimilating $CO_2$ concentration, so as to provide a robust integrated estimate of net carbon exchange and its component processes (photosynthesis and respiration) over Arctic and boreal regions.

## Methods

**Summer $CO_2$ drawdown from the Barrow $CO_2$ data.** We used daily records of in situ atmospheric $CO_2$ concentration at Barrow (71°N, 157°W, Alaska) for 1979–2012 archived by the National Oceanic and Atmospheric Administration (NOAA) Earth System Research Laboratory[25]. We obtained the detrended seasonal $CO_2$ curve by separating the seasonal cycle from the long-term trend and short-term variations, fitting a function consisting of a quadratic polynomial for the long-term trend and four harmonics for the seasonal cycle to the daily data and then digitally filtering the residuals from the fitted function[17,45]. A 1.5-month and a 390-day full-width half-maximum-value averaging filter were used for the digital filtering of residuals to remove the short-term variations and the long-term trend, respectively. Some unreliable $CO_2$ observations can strongly affect estimates of the seasonal cycle and therefore the net summer uptake, so any data lying outside five standard deviations of the residuals between the original data and the fit were discarded from the original daily time series[17]. We calculated summer $CO_2$ drawdown (SCD), which was adopted as an indicator of net summer $CO_2$ uptake in three ways. For the main analysis, SCD was calculated as the difference of $CO_2$ concentration between the first week of July and the last week of August. For the robustness tests, SCD was calculated as the difference of $CO_2$ concentration

between the climatological day of the year when $CO_2$ crossed its annual mean level (the climatological spring zero-crossing date) and the climatological day of the year of minimum atmospheric $CO_2$ concentration (the climatological trough date). We found that the climatological spring zero-crossing date at Barrow was around the 178th day of the year and the climatological trough date was around the 235th day of the year using the detrended seasonal $CO_2$ curve for 1979–2012 (Supplementary Fig. 1). The spring zero-crossing and trough dates could vary across years, so we also calculated SCD for testing the robustness using inter-annually varying values of the spring zero-crossing and trough dates. In addition, we used weekly records of atmospheric $CO_2$ concentration from the NOAA Earth System Research Laboratory[46] at Barrow to derive net summer $CO_2$ uptake. The same methods of fitting functions and digital filtering as used for the daily data were used for the weekly $CO_2$ data to obtain a detrended seasonal $CO_2$ curve; the only exception was that we did not need to remove the potential outliers, because they had already been excluded in the GLOBVIEW–$CO_2$ data.

**Area of the summer $CO_2$ footprint for Barrow.** The change in $CO_2$ concentration at Barrow is driven mainly by carbon exchanges in northern terrestrial ecosystems[47]. We used two methods to define the specific regions that constitute the area of the summer footprint. We first used the Lagrangian particle dispersion model FLEXPART (version 8.2)[31] with a driving wind field obtained from the European Centre for Medium-Range Weather Forecasts (ECMWF). FLEXPART simulations were available for 1985–2009 at a time resolution of three hours. A map of Potential Emission Sensitivity for Barrow was calculated by simulating the backwards-in-time transport of 40,000 infinitesimal air 'particles' for 20 days and integrating over time at the lowest model output layer (0–100 m) over the 20 days[48]. We then averaged the Potential Emission Sensitivity for July and August for each year during 1985–2009 to obtain an average summer footprint area for each year.

The second method used to define the summer footprint area was based on the adjoint code of the LMDZ atmospheric transport model[49]. The adjoint code calculates the partial derivatives of concentration measurements for fluxes at any time before a given $CO_2$ observation. We calculated the partial derivatives for the mean $CO_2$ measurements made at Barrow during the last week of August for fluxes at daily resolutions since the start of July, for all years of this study. Integrating the derivatives for a given number of days (Supplementary Fig. 5) within 1 year quantified the change in $CO_2$ concentration at Barrow per unit change in flux (kg C m$^{-2}$ h$^{-1}$) for all days before the measurement. We then averaged the derivatives for each year between 1979 and 2012 to obtain a multi-year mean area of the summer footprint.

**Climatic and satellite-derived products.** We used monthly climatic data (temperature, precipitation, and cloud cover) at a spatial resolution of 0.5° for 1901–2012 from the Climate Research Unit, University of East Anglia (CRU TS4.0 dataset)[50]. Climatic observations at high latitudes are rare, so we also applied another climatic dataset, WATCH (WATer and global CHange) Forcing Data, to the ERA-Interim data (http://www.eu-watch.org/gfx_content/documents/README-WFDEI.pdf)[51].

Satellite-derived NDVI, used as a proxy for vegetation productivity, was retrieved from the third-generation of the Advanced Very High Resolution Radiometer (AVHRR) developed by the Global Inventory Modeling and Mapping Studies (GIMMS) group[26]. The GIMMS NDVI for 1981–2014 has a spatial resolution of one-twelfth of a degree (~8 km) and a 15-day interval (available at https://ecocast.arc.nasa.gov/data/pub/gimms/3g.v1/). Solar-induced chlorophyll fluorescence (SIF) that reflects photosynthetic signals from the molecular origin is increasingly used as a physiological-based proxy for gross primary productivity (GPP)[52–55]. Here we used an extended satellite-derived SIF dataset for the period 2001–2012[35]. This dataset is generated by a trained neural network that integrates surface reflectance from the MODerate-resolution Imaging Spectroradiometer (MODIS) and SIF from the Orbiting Carbon Observatory-2 (OCO-2). The extended SIF dataset, which is available for the all-sky condition, has a spatial resolution of 0.05 degrees and a temporal resolution of 4 days. Two other terrestrial productivity products were also used. The first was a 30-year global dataset of satellite-derived net primary productivity (NPP), which was calculated using the Moderate Resolution Imaging Spectroradiometer (MODIS) NPP algorithm driven by 30-year (1982–2011) GIMMS FPAR (the fraction of photosynthetically active radiation absorbed by the vegetation) and data for leaf area index (LAI)[27]. The second was the GPP product (0.5° × 0.5°, monthly, 1982–2011) generated by integrating a global set of eddy covariance sites, satellite remote sensing, and meteorological data in a machine-learning algorithm[28]. This global product has been widely adopted to aid the understanding of the spatiotemporal dynamics of the global carbon cycle[56] and to benchmark process-based terrestrial models[28,57,58]. In addition, we used monthly data for root-zone soil moisture and evapotranspiration from the Global Land Evaporation Amsterdam Model (GLEAM) version 3.0a dataset at a spatial resolution of 0.25° × 0.25°[37]. This dataset assimilates microwave observations of surface-soil moisture from the European Space Agency (ESA)–Climate Change Initiative (ESA–CCI) dataset in a multi-layer water-balance module[38]. We also used the sea-ice extent data that are derived from the Sea Ice Index Version 3 dataset (http://nsidc.org/data/G02135).

**Atmospheric $CO_2$ inversion system**. We gathered estimates of monthly net biome production (NBP) from MACC (Monitoring Atmospheric Composition and Climate) (version v14r2, http://copernicus-atmosphere.eu/) for 1979–2011[49]. The MACC $CO_2$ atmospheric-inversion system, based on the global tracer transport model LMDZ[29], adopts a variational inversion formulation to estimate optimized surface $CO_2$ fluxes from nearly 130 $CO_2$ observing stations, with a spatial resolution of 3.75° × 1.875° and a temporal resolution of 8 days, separately for daytime and nighttime.

**Simulations of the terrestrial ecosystem models**. We used simulated net primary productivity and heterotrophic respiration from nine process-based ecosystem models: Community land Model Version 4.5 (CLM4.5), Integrated Science Assessment Model (ISAM), the Joint UK Land Environment Simulator (JULES), Lund-Potsdam-Jena DGVM (LPJ), Lund-Postam-Jena General Ecosystem Simulator (LPJ-GUESS), the Land surface Processes and eXchanges (LPX-Bern), O-CN, Organizing Carbon and Hydrology In Dynamic Ecosystems (ORCHIDEE) and the Vegetation Integrative Simulator for Trace gases (VISIT). All models followed the protocol described by the historical climatic carbon-cycle model comparison project (Trendy) (http://dgvm.ceh.ac.uk/files/Trendy_protocol%20_Nov2011_0.pdf). All models used prescribed static vegetation maps. Each model was run from its pre-industrial equilibrium (assumed at the beginning of the 1900s) to 2012 and was forced by both observed historical climate changes and rising $CO_2$ concentrations.

**The linkage between atmospheric $CO_2$ and surface $CO_2$ fluxes**. Atmospheric $CO_2$ concentration and net carbon fluxes are two different parameters: atmospheric $CO_2$ concentrations integrate net carbon fluxes from different regions through atmospheric transport. To quantify the linkage between atmospheric $CO_2$ concentration and net surface $CO_2$ fluxes, we use the LMDZ atmospheric transport model to convert net $CO_2$ fluxes into an estimate of atmospheric $CO_2$ concentration at Barrow station. The transport model is nudged with winds from the European Centre for Medium-Range Weather Forecasts (ECMWF) reanalysis. We performed an ensemble of transport simulations for the period 1979–2012, using different NBP simulations from the terrestrial ecosystem models participating in the Trendy project and the NBP dataset from the MACC inversion system. Our results showed that simulated changes in SCD from the period 1996–2012 to 1979–1995 at Barrow station are strongly correlated with their corresponding summer NBP changes north of 50°N ($R^2 = 0.67$, $P < 0.01$) (Supplementary Fig. 21). Although atmospheric $CO_2$ concentration is not a direct measure of terrestrial net $CO_2$ fluxes, change in SCD at the Barrow station is a good indicator of change in net surface $CO_2$ fluxes north of 50°N.

We performed further transport simulations to estimate the linkage between SCD and net surface $CO_2$ fluxes based on perturbations of the fluxes. Specifically, for a given summer NBP map, we performed both control and perturbed simulations using the LMDZ transport model. In the perturbed case, we increased summer NBP by a scaling value so that summer NBP north of 50°N increased by 1 PgC for each year of the period 1979–2012. For a given year, the scaling value for each pixel north of 50°N is calculated as the ratio of summer NBP to one unit of PgC. The sensitivity of SCD change to NBP change is then computed as the difference in SCD between the control and perturbed simulations, relative to one unit increase of PgC in terrestrial NBP north of 50°N.

To understand the uncertainty related to the choice of different NBP maps, we considered different modeled NBP from the Trendy project and the NBP from the MACC inversion system. Our results show that, on average, one unit increase of PgC in terrestrial NBP north of 50°N can lead to an increase of 5.49 ± 2.70 ppm in SCD at the Barrow station (Supplementary Fig. 22). Note that averaging an ensemble of modeled results does not necessarily give the correct sensitivity. The accurate estimate of this sensitivity relies on the accuracies of the estimates of net surface $CO_2$ fluxes and the atmospheric transport model.

**Statistical analyses**. We quantified the decadal change in interannual correlations between summer (July and August) temperature and SCD by calculating partial-correlation coefficients between summer temperature and SCD ($R_{SCD-T}$) and controlling for the effects of summer precipitation and cloud cover (all variables detrended) during the earlier (1979–1995) and later (1996–2012) periods by randomly selecting 14 years from the corresponding period. Before performing the correlation analyses, we calculated the standard deviation of daily temperature between July and August for each year and found no significant change during the entire period (1979–2012) (Supplementary Fig. 23), suggesting that the temperature variance is relative stable. We also estimated the interannual sensitivity of SCD to summer temperature ($\gamma_{SCD-T}$) for each period as the slope of a multiple regression line of SCD against summer temperature, precipitation, and cloud cover (all variables detrended). A two-sample $t$-test was used to determine the significance of the differences in $R_{SCD-T}$ ($\gamma_{SCD-T}$) between the earlier and later periods. We also calculated the partial-correlation coefficients linking temperature to satellite-derived NDVI ($R_{NDVI-T}$), satellite-derived SIF ($R_{SIF-T}$), satellite-derived NPP ($R_{NPP-T}$), and GPP based on flux-tower data ($R_{GPP-T}$) using a similar method. The NDVI data began in 1982, so we randomly selected 11 years from 1982 to 1995 to calculate the frequency distribution of $R_{NDVI-T}$. All these analyses were first performed for north of 50°N and then for each pixel.

All climatic variables for the main analysis were calculated as spatial averages over the vegetated land north of 50°N. Vegetated land was defined as areas with a mean annual NDVI > 0.1 for 1982–2012. We also calculated the spatial averages of the climatic variables, as part of the robustness tests, by weighting the calculated sensitivities (potential emission sensitivity from FLEXPART and surface flux sensitivity from LMDZ) of different vegetated land areas within the multiple-year mean summer-footprint area. We did not set a cutoff value of sensitivities to select the summer footprint for the weighting, because a large fraction of the land surface had low sensitivity values. For example, values of potential emission sensitivity from FLEXPART above 0.1 s constituted 93.35% of the land-surface signal, and values above 1 s constituted 41.70% of the land-surface signal.

## Data availability

The authors declare that ORCHIDEE-LMDZ transport simulation results are available from the corresponding author upon request. All other data supporting the findings of this study are available within the paper.

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

## Acknowledgements

This study was supported by the Strategic Priority Research Program (A) of the Chinese Academy of Sciences (grant XDA20050101, XDA19070303), the National Natural Science Foundation of China (41530528, 41871104), the 13th Five-year Informatization Plan of Chinese Academy of Sciences (Grant XXH13505-06) and the Thousand Youth Talents Plan project in China. P.C., I.J., and J.P. were funded by the European Research Council Synergy grant SyG-2013-610028 IMBALANCE-P. We thank the TRENDY modelling group for providing the model simulation data.

## Author contributions

T.W. and S.L.P. designed the research, D.L. performed statistical analyses based on the Barrow $CO_2$ observations and gridded carbon fluxes, Y.L.W. and J.B. conducted the footprint analysis, T.W. drafted the paper, D.L., S.L.P., Y.L.W., J.B., P.C., J.P. and I.J. contributed to the writing, X.Y.W., H.G., X.L., M.H., Y.L., Y.W.L., S.S.P., H.Y., Y.T.Y., Y.Y. and Y.T.Z. contributed to the interpretation of the results.
