## [Peer Review File · Nature Communications]

Reviewers' comments:

Reviewer #1 (Remarks to the Author):

Summary:

In this analysis of “Emerging negative warming impact on summer carbon uptake in northern ecosystems” Wang et al. focus on summer carbon uptake at high latitudes and its changing relationship with temperature. Clearly the behavior of the global C cycle is changing over time and nowhere is this more evident than in the arctic where recent research has revisited changes in the seasonal cycle first identified by Keeling et al. 1996. Here Wang et al. conclude that summer C uptake is negatively correlated with summer average temperature and becoming increasingly more significant over time. While this analysis focusing on the peak of summer uptake is somewhat novel and potentially shows a decreased enhancement of photosynthesis due to temperature at high latitude, I have some concerns about assumptions made in analyzing this timeseries that is clearly changing through time.

General Comments:

1.) By applying a fixed time window to their analysis the authors assume that the seasonal cycle has been stationary and that no phase shift has occurred in the timeseries. However, we know that the phase of the seasonal cycle at Barrow has shifted by 0.2 days/yr (Graven et al. 2013). Although this seems fairly trivial, this leads to a 15 day advance in the phase over a 30 year timeseries- thus in the Arctic June is the new July! Previous analyses have used the maxima and minima in the timeseries to estimate peak to trough amplitudes, instead of this fixed time window of summer carbon uptake. Therefore it is conceivable that Wang et al's calculation of summer C drawdown (SCD) has gone down simply as a result of the phase change (see Fig2):

Figure 1. Revised from Wang et al's supplementary figure 1. Here the red line illustrates how the change in the phase as the seasons advance could lead to the false impression that SCD is going down.

This may not affect their analysis, but I don't think that it can be neglected. I am not sure exactly how to consider this in their analysis, but perhaps looking at the difference between the zero-crossing and the minima is a better approach than looking at a fixed window over time. Of course this also presents a problem when looking at the fixed window of temperature data in your regression analysis. Further this analysis seems to assume that the temperature variance is stationary through time by looking at the 90th percentile, but it is not shown that the variance is constant.

2.) It is interesting that there is such a renewed focus on the seasonal cycle originally identified by Keeling et al. in 1996 and yet nobody has attempted to solve the most intriguing puzzle identified by Keeling et al. that there was a strong positive relationship with temperature-**but that seasonal amplitude lagged temperature by 1 to 2 years!** Given the strong interannual variability in Fig. 1 of Wang et al. I think that if they did a lagged temperature analysis they would probably find a positive correlation with temperature in the preceding years. While Keeling did not provide a conclusive explanation of this time lag it suggests complex feedback mechanisms with some memory effects. This previous relationships should at least be mentioned in the discussion.

3.) Barrow is located on the Arctic Ocean and is also influenced by potential changes in air-sea gas exchange. Figure S2 is misleading and suggests that this site is only influenced by land fluxes. While it may be safe to assume that air-sea gas exchange has very little impact on the seasonal cycle of atmospheric CO₂ at other locations, this is probably not valid due to 'sea-ice phenology' that is changing over time.

Specific Comments:

L 38 This is true except for all the papers that have focused on changes in the amplitude of the seasonal cycle!

L 60 this fixed window ignores phase changes in the timeseries (see Graven et al.)

L 61 I don't know why it is becoming popular to summarize results in the introduction because I just read a summary of the results in the abstract. This section is redundant.

L 86 These changes in gamma results should be presented in Fig. 1 (or else in supplementary).

L 104 No arctic ocean in the footprint- impossible.

L 116 It sounds as though Wang et al. at least considered the phase shift in their analysis. However the caption to Figure S3 still implies a fixed window from DOY 178 to 235. What if you let this window vary based on the data. These results are possibly more important. Did they shift the temperature window accordingly in their regression analysis?

L 162 This assumes that temperature variance is not changing over time-not a valid assumption.

L 177 this is interesting because increased soil moisture could promote CH₄ instead of CO₂ losses

L 183 I would restructure discussion and start with total ecosystem respiration (TER) and then move towards heterotrophic respiration (HR).

L 208 I suggest you go back and re-read the Keeling paper and discuss temperature sensitivity in a broader context

Assume that air sea gas exchange plays little role in changing C uptake over time. (Fig S2)

Reviewer #2 (Remarks to the Author):

This is an interesting paper containing original analyses and that is potentially worthy of publication. The data, data analysis, and central conclusions are interesting, seem solid, and seem eventually publishable.

However, the paper needs major rewriting for clarity, to accurately report the intent and results, for grammar, the literature, and for accuracy.

Much more care needs to be given to insuring that each word and phrase portrays the situation accurately, clearly, and that what is meant is accurately portrayed. Non-careful writing has led to many inaccurate statements because the intent of the writing is not accurately or appropriately conveyed.

I suggest that some of the fluent English language authors, or an editor, take a more active role in the re-writing and editing of the manuscript. Much more work is needed before this manuscript is ready for publication.

Below I give some examples. This list is not exhaustive, rather it is meant to point out the type of problems that exist with the manuscript. If only the items below are fixed, and that the sense of this review is not conveyed to the entire manuscript, it will still not be ready for publication.

Specific comments:

L20-21. As written, this is a ridiculous statement. It shows non-careful formulation, a lack of specificity, and possibly a lack of knowledge of the literature. Certainly, most of the previous high latitude carbon cycle have been done in the summer. The major lack is information on the fall, winter, and spring periods. See work done at NARL in the '60s and during IBP for example and early eddy covariance work in the Arctic. The authors may mean a particular type of inversion study, but they do not say that.

L27. Why surprising? Change "surprising" to "however".

L27-32. Awkward, confusing, unclear what is meant. Rewrite carefully and clearly.

L33. What type of disturbances? How does temperature cause Arctic disturbances?

L33-34. This is debated. There are many reports of many northern areas switching from a C sink to an annual C source in recent decades. Review the literature starting in the 1990s (see e.g. Oechel et al. Nature 1993, 2000 and 2014 to cite a few).

L36, 39 Here and elsewhere. Mostly, what is discussed is C fluxes not C cycles. Use C fluxes.

L38. This is not true. You are cherry picking the results and papers cited.

L39. This is simply not true. The greatest amount of attention has been given to summer. This is a ridiculous assertion. If you mean a particular type of study, then say so.

L42-44. Not true. The spring has ideal climatic conditions for C uptake. It is just that the plants have not developed yet. Phenology may be controlling spring/early summer uptake.

L 48. Unclear what is meant by the "response of the carbon cycle". What is "the" carbon cycle?

L50. Simply not true. The summer carbon cycle is the best studied of all seasons. See results from the Arctic IBP from the early 1970s and following studies.

L93. What does "Main" summer footprint mean? 50% of the entire foot print? Be specific.

L109. "does not affect" should be "does affect"

L202. "profoundly". No profound alteration has been demonstrated in this paper. Remove "profoundly".

L203. Shifting to a significantly "negative state" is unclear. Rewrite the sentence.

L206 Change to: to "increased" net carbon..."

L210-213. Cite papers going back to IBP (1970s) and since including those in nature.

Fig 3. Figure hard to interpret. Add variable "+" and "-" to indicate varying positive and negative effects.

L243. Performed "CO2 'measurement' analysis" is unclear. What is measurement analysis? Explain.

L306-309. What tower observations? How was this done? Where are the results?

L354-355. What does this mean? How do you know this? How do you know that summer CO2 conc. Observed at BRW are mainly from terrestrial sources within the summer footprint area. Reference the SI and explain. How did you rule out ocean sources or sinks?

Reviewer #3 (Remarks to the Author):

The manuscript demonstrates a significant change in the interannual relationship between summer land temperature and summer drawdown of atmospheric CO₂ at high northern latitudes over recent decades. Careful comparison of the changing correlations between temperature and CO₂ drawdown with proxy data for photosynthesis and ecosystem respiration indicate that the change in temperature response most likely results from heat and drought stress on plant productivity as a result of an increase in very warm weather. The data are from well-established measurement programs, the analysis is straightforward, the implications are important, and the paper is very well-written. I recommend publication with minor revisions as explained below.

The authors use a backward-in-time Lagrangian particle transport model (FLEXPART) to calculate upstream "concentration footprints" for the CO₂ measurements at Point Barrow. The summertime footprint is used to define averaging areas for the various proxy data (NDVI, NPP, etc) that are used to investigate hypothetical mechanisms that might explain the change in temperature response of CO₂ drawdown. This is a reasonable method and is based on well-established meteorological data and models, but it's surprising that the resulting footprint was localized to Alaska and eastern Siberia.

The authors cite studies by Gray et al (2014) and Zeng et al (2014) both of which attribute changes in high-latitude CO₂ seasonality to increases in agricultural production in midlatitudes. A recent analysis by Barnes et al (2016) showed that CO₂ seasonality at Point Barrow is strongly influenced by ecosystem processes in temperate latitudes, and that transport by baroclinic waves carries this seasonality into the high Arctic.

I suspect that the overall conclusions of the present manuscript do not depend sensitively on the footprint over which the proxy data are averaged. It seems likely that temperate ecosystems may

also show changes in temperature response with recent warming. But the authors should address the issue of long-distance transport of seasonal CO₂ variations from lower latitudes.

How sensitive are their conclusions to the particular footprint calculations they used? Perhaps 20 days of back-trajectory calculations are insufficient to track CO₂ backward from Point Barrow to temperate croplands, pastures, and forests? At the very least the manuscript should cite recent work on long-distance meridional transport of CO₂ seasonality and consider the implications for their results.

Reference

Barnes, E. A., N. Parazoo, C. Orbe, and A. S. Denning (2016), Isentropic transport and the seasonal cycle amplitude of CO₂, *J. Geophys. Res. Atmos.*, 121(13), 8106–8124, doi: 10.1002/2016JD025109.

To Reviewer #1

[General Comment] *In this analysis of "Emerging negative warming impact on summer carbon uptake in northern ecosystems" Wang et al. focus on summer carbon uptake at high latitudes and its changing relationship with temperature. Clearly the behavior of the global C cycle is changing over time and nowhere is this more evident than in the arctic where recent research has revisited changes in the seasonal cycle first identified by Keeling et al. 1996. Here Wang et al. conclude that summer C uptake is negatively correlated with summer average temperature and becoming increasingly more significant over time. While this analysis focusing on the peak of summer uptake is somewhat novel and potentially shows a decreased enhancement of photosynthesis due to temperature at high latitude, I have some concerns about assumptions made in analyzing this time series that is clearly changing through time.*

[Response] Many thanks for your comments that help us to improve our MS. We carefully revised our manuscript following your comments and suggestions.

[Comment 1] *By applying a fixed time window to their analysis the authors assume that the seasonal cycle has been stationary and that no phase shift has occurred in the time series. However, we know that the phase of the seasonal cycle at Barrow has shifted by 0.2 days/yr (Graven et al. 2013). Although this seems fairly trivial, this leads to a 15 day advance in the phase over a 30 year time series- thus in the Arctic June is the new July! Previous analyses have used the maxima and minima in the time series to estimate peak to trough amplitudes, instead of this fixed time window of summer carbon*

uptake. Therefore, it is conceivable that Wang et al's calculation of summer C drawdown (SCD) has gone down simply as a result of the phase change (see Fig2):

Figure 1. Revised from Wang et al's supplementary figure 1. Here the red line illustrates how the change in the phase as the seasons advance could lead to the false impression that SCD is going down.

This may not affect their analysis, but I don't think that it can be neglected. I am not sure exactly how to consider this in their analysis, but perhaps looking at the difference between the zero-crossing and the minima is a better approach than looking at a fixed window over time. Of course this also presents a problem when looking at the fixed window of temperature data in your regression analysis. Further this analysis seems to assume that the temperature variance is stationary through time by looking at the 90th percentile, but it is not shown that the variance is constant.

[Response] Following your suggestion, we performed the analysis using an inter-annually varying window approach, in which the summer CO₂ drawdown (SCD) is defined as the difference in CO₂ concentration between the climatological day of the year when the detrended CO₂ concentration crossed its long-term mean downwards

(climatological spring zero-crossing date) and the climatological day of the year when detrended CO₂ concentration reached its annual minimum (climatological minimum date) (Figure R1). Accordingly, the climate variables (including temperature, precipitation and cloudiness) for each year are averaged between inter-annual varying spring zero-crossing date and inter-annual varying minimum date across ecosystems north of 50°N.

As shown in Figure R1, both spring zero-crossing date and the minimum date advance at rates of 0.22 and 0.21 day yr⁻¹, respectively. The SCD increases at a rate of 0.065 ppm yr⁻¹ and the corresponding temperature between zero-crossing date and minimum date increases at a rate of 0.04 °C yr⁻¹. By calculating the partial-correlation between SCD and temperature in the two periods, we again found that the correlation is not significant in the earlier 17 years (1979–1995) but is negatively significant in the recent 17 years (1996–2012) (Figure R2). This result is in line with that based on the fixed-window approach, therefore suggesting that the varying phase of CO₂ seasonal cycle should not affect our main findings.

In order to test whether the temperature variance is stationary through time, we calculated the standard variation of daily temperature between July and August for each year. Our results indicated that there is no significant trend ($P = 0.5$) in the standard variation of summer temperature during the entire period (1979–2012) (Figure R3), suggesting that the stationarity of temperature variance should not be

regarded as an influencing factor in explaining the recent intensified negative temperature impact on summer carbon uptake. This result is also found if the standard deviation is calculated from summer temperature anomalies that are obtained by removing the low frequency signal from the daily temperature series based on the Empirical Mode Decomposition (Figure R4).

In order to resolve the reviewer's concerns, we have changed "*Second, to test whether this shift is due to the use of the month of August that includes a short time period during which CO₂ uptake declines (Supplementary Fig. 1), we also computed SCD as the difference of CO₂ concentrations between the day of the year when detrended CO₂ crosses down its annual mean level and that when detrended CO₂ reaches its annual minimum. Again, we found a similar shift in R_{SCD-T} from -0.28 ($P = 0.32$) during 1979–1995 to -0.67 ($P < 0.01$) during 1996–2012 (Supplementary Fig. 3).*" into "*We also explored the robustness of the result to the method used to calculate SCD. SCD was initially calculated as the difference in the CO₂ concentration between the first week of July and the last week of August in the detrended CO₂ record, but CO₂ uptake decreased briefly in August (Supplementary Fig. 1), which could conceivably impact the results. Therefore, we calculated SCD using alternative methods. We first set SCD as the difference in CO₂ concentration between the climatological day of the year when the detrended CO₂ concentration crossed its long-term mean downwards (climatological spring zero-crossing date) and the climatological day of the year when detrended CO₂ concentration reached its annual minimum (climatological trough date). R_{SCD-T}*

calculated from this different definition of SCD varied from -0.28 ($P = 0.32$) during 1979–1995 to -0.67 ($P < 0.01$) during 1996–2012 (Supplementary Fig. 10), similar to the original SCD definition. The spring zero-crossing date and the trough date for 1979–2012 both advanced at rates of 0.22 and 0.21 day year⁻¹, respectively (Supplementary Fig. 11), so a fixed window for defining SCD using the climatological zero-crossing and trough dates may not be appropriate. We thus performed an additional analysis that allowed the spring zero-crossing and trough dates to vary interannually. We again found that R_{SCD-T} shifted, from 0.42 ($P = 0.12$) during 1979–1995 to -0.52 ($P < 0.01$) during 1996–2012 in that new test (Supplementary Fig. 12). From this, one can conclude that the detected shift in R_{SCD-T} is not an artifact of the method used to define SCD.”. (Line 135–153 on page 7–8)

In the Methods, we also changed “*We computed summer CO₂ drawdown (SCD), which is adopted as an indicator of net summer CO₂ uptake, as the difference of CO₂ concentration between the first week of July and the last week of August. In addition, based on the detrended seasonal CO₂ curve, the mean estimate of minimum atmospheric CO₂ concentration and the day of year when CO₂ crosses down through its annual mean level during the period 1979–2012 at Barrow are around the day of the year (DOY) 178 and 235 (Supplementary Fig. 1). Therefore, we also calculated SCD as the difference of CO₂ concentration between DOY 178 and DOY 235, which can effectively represent a period of strong CO₂ drawdown from the available CO₂ record.”* into “*We calculated summer CO₂ drawdown (SCD), which was adopted as an indicator*

of net summer CO₂ uptake in three ways. For the main analysis, SCD was calculated as the difference of CO₂ concentration between the first week of July and the last week of August. For the robustness tests, SCD was calculated as the difference of CO₂ concentration between the climatological day of the year when CO₂ crossed its annual mean level (the climatological spring zero-crossing date) and the climatological day of the year of minimum atmospheric CO₂ concentration (the climatological trough date). We found that the climatological spring zero-crossing date at Barrow was around day of the year 178 and the climatological trough date was around day of the year 235 using the detrended seasonal CO₂ curve for 1979–2012 (Supplementary Fig. 1). The spring zero-crossing and trough dates could vary across years, so we also calculated SCD for testing the robustness using interannually varying values of the spring zero-crossing and trough dates.”. (Line 332–341 on page 18)

In addition, we have added the following text into Methods part.

“Before performing the correlation analyses, we calculated the standard deviation of daily temperature between July and August for each year and found no significant change during the entire period (1979–2012) (Supplementary Fig. 19), suggesting that the temperature variance is relative stable.”. (Line 430–433 on page 22–23)

Figure R1. The time series of spring zero-crossing date (a), the trough date (b), the SCD defined by inter-annually varying zero-crossing date and trough date (c) and the average temperature between inter-annually varying zero-crossing date and trough date (d).

Figure R2. Same as Figure 1, but using SCD and corresponding climate calculated for the period between inter-annually varying date when CO₂ crosses down zero (inter-annually varying spring zero-crossing date) and inter-annually varying date

when CO₂ reaches its annual minimum (inter-annually varying trough date).

Figure R3. The time series of the standard variation of daily temperature between July and August during the period 1979–2012.

Figure R4. The time series of the standard variation of daily temperature between July and August during the period 1979–2012. The Empirical Mode Decomposition analysis is adopted to decompose the daily temperature series to multiple frequencies. We only contain the high frequency signals with the Hurst exponent lower than 0.5.

[Comment 2] *It is interesting that there is such a renewed focus on the seasonal cycle originally identified by Keeling et al. in 1996 and yet nobody has attempted to solve the most intriguing puzzle identified by Keeling et al. that there was a strong positive*

relationship with temperature-but that seasonal amplitude lagged temperature by 1 to 2 years! Given the strong interannual variability in Fig. 1 of Wang et al. I think that if they did a lagged temperature analysis they would probably find a positive correlation with temperature in the preceding years. While Keeling did not provide a conclusive explanation of this time lag it suggests complex feedback mechanisms with some memory effects. This previous relationship should at least be mentioned in the discussion.

[Response] Here we follow the reviewer's suggestion to conduct the lagged correlation analysis between SCD and temperature at the inter-annual timescale (Figure R5). In order to well resolve the reviewer's concern, we have added the following discussion into the revised MS.

“The amplitude of the seasonal CO₂ concentration at Barrow was found to lag behind temperature by about two years³⁰, and it was suggested that this lag was due to a lag in the response of net primary production to temperature. In contrast, SCD was not significantly correlated with summer temperature in the previous two years (or one year) for any of the study periods (1979–2012, or the two periods 1979–1995 and 1996–2012). This lack of lagged-correlation coincides with a non-significant lagged-response of summer productivity to temperature in the previous one or two years (Supplementary Fig. 3). This result does not contradict the result from Keeling et al. (1996)³⁰ but it shows that if there is a lagged response of the peak-to-peak CO₂ amplitude to temperature, it is not due to a lag of summer CO₂ uptake.”. (Line 94–103 on page 5)

Figure R5. The inter-annual partial-correlation of summer CO₂ drawdown (SCD) (a), gross primary productivity (GPP) (b) and net primary productivity (NPP) (c) with summer temperature at various time lags (zero-year, previous one year and previous two years) for 1982–2011 (black), 1979–1995 (blue) and 1996–2012 (red) periods. °, * and ** indicates that partial-correlation coefficient is statistically significant at $P < 0.1$, $P < 0.05$ and $P < 0.01$, respectively.

[Comment 3] *Barrow is located on the Arctic Ocean and is also influenced by potential changes in air-sea gas exchange. Figure S2 is misleading and suggests that this site is only influenced by land fluxes. While it may be safe to assume that air-sea gas exchange has very little impact on the seasonal cycle of atmospheric CO₂ at other locations, this is probably not valid due to ‘sea-ice phenology’ that is changing over time.*

[Response] We agree with the reviewer that we need to consider the potential impact of changes in air-sea gas exchanges, since Point Barrow is located on the edge of the

Arctic Ocean and its CO₂ variation could be influenced by ocean carbon fluxes according to our updated summer footprint area (Figure R6).

As stated by the reviewer, changes in sea-ice phenology (melt onset and freeze-up) could induce variations in air-sea gas exchanges, potentially contributing to the variation of CO₂ concentration observed at Barrow station. In the context of global warming, the date of the onset of melting of Arctic sea ice advanced at rates around 2.8 day decade⁻¹ during 1979–2004 (Markus et al., 2009), which might stimulate summer air-sea CO₂ flux in the Arctic and potentially contribute to the variations of summer CO₂ concentration at Barrow. To test this hypothesis, we used sea surface temperature since which is closely correlated with percentage of open water in the Arctic during the summer (Galbraith and Larouche, 2011) and can then be regarded as an indicator for melt onset date. Our analysis shows that the correlation between SCD and Arctic SST north of 50°N are not significant in both periods (Figure R7). A possible indirect effect of SSTs on the shift in R_{SCD-T} was investigated using the partial-correlation between SCD and land temperature after controlling for the effects of cloudiness, precipitation and Arctic SSTs. Our results show that there is a similar shift in R_{SCD-T} from 0.51 ($P = 0.07$) during 1979–1995 to -0.67 ($P < 0.01$) during 1996–2012 (Figure R8), tentatively suggesting the limited impact of changes in air-sea exchanges.

In order to address the reviewer's concern, we have updated Figure S4 to include potential emission sensitivity (PES) in ocean area. Furthermore, we have also added

the following paragraph in the revised MS.

“We further investigated whether the shift in R_{SCD-T} may have been indirectly related to changes in the onset date of the Arctic sea-ice melt. The date of the onset of melting of sea ice advanced at rates around $2.8 \text{ day decade}^{-1}$ during 1979–2004 at Chukchi and Beaufort Seas³². Earlier melting could increase the air-sea CO_2 flux in the summer and potentially contribute to the variations of summer CO_2 concentration at Barrow. We would then expect a correlation between SCD and Arctic sea-surface temperature (SST), because SST is closely correlated with the percentage of open water in the Arctic during the summer³³. We tested this hypothesis by calculating the correlation between SCD and summer (July-August) SSTs north of 50°N for the two study periods but found no significant correlation for either period (Supplementary Fig. 13). A possible indirect effect of SSTs on the shift in R_{SCD-T} was investigated using the partial-correlation between SCD and land temperature after controlling for the effects of cloudiness, precipitation and Arctic SSTs. R_{SCD-T} shifted similarly to the original calculation (Supplementary Fig. 14), suggesting that the impact of changes in air-sea exchanges on the SCD changes was limited.”. (Line 155–168 on page 8)

Figure R6. Summer footprint area for Barrow derived from FLEXPART Lagrangian particle dispersion model during three time periods. The simulations were only available from 1985 to 2009.

Figure R7. Correlation between summer CO₂ uptake at Barrow and SST north of

50°N. The lines are time series of anomaly of summer CO₂ drawdown (SCD, black) and SST (red) calculated as the average for July and August across ocean north of 50°N. The inset illustrates the interannual simple correlation coefficient between SCD and Arctic SSTs.

Figure R8. Same as Figure R7, but calculated partial-correlation between SCD and air temperature with controlling precipitation, cloudiness and Arctic SSTs. The lines are time series of anomaly of summer CO₂ drawdown (SCD, black) and temperature (red) calculated as the average for July and August across ocean north of 50°N. The inset illustrates the interannual partial correlation coefficient between SCD and Arctic SSTs. * and ** indicates that partial correlation coefficient is statistically significant at $P < 0.05$ and $P < 0.01$, respectively.

[Specific Comments]

[Comment 4] L 38 *This is true except for all the papers that have focused on changes in the amplitude of the seasonal cycle!*

[Response] We changed this sentence to: “*Numerous field studies in high-latitude ecosystems have been conducted in the summer¹⁰⁻¹⁸, but the results of these local experiments are limited in extent and are difficult to scale up to larger regions. Observations of atmospheric CO₂ concentration from atmospheric stations at high latitudes provide complementary monitoring of the dynamics of carbon exchange in northern ecosystems. But most of these studies have concentrated on the amplitude of the seasonal cycle or the spring and autumn boundaries of the growing season¹⁻³, little attention has been given to the summer.*”. (Line 42–48 on page 3)

[Comment 5] *L 60 this fixed window ignores phase changes in the time-series (see Graven et al.)*

[Response] Following your constructive suggestion, we have added an additional analysis using an inter-annually varying window approach, in which the summer CO₂ drawdown (SCD) is defined as the difference of CO₂ concentration between the date when the detrended CO₂ concentration crossed its long-term mean downwards (spring zero-crossing date) and the climatological day of the year when detrended CO₂ concentration reached its annual minimum (trough date). Our analysis indicated that the results are in line with that based on the fixed-window approach, suggesting that the varying phase of CO₂ seasonal cycle should not affect our main findings. The details can be seen in the response of **Comment 1** by Reviewer #1.

[Comment 6] *L 61 I don't know why it is becoming popular to summarize results in the*

introduction because I just read a summary of the results in the abstract. This section is redundant.

[Response] We have changed “*Here we analyzed the long-term record of atmospheric CO₂ concentrations from the Point Barrow (BRW) atmospheric CO₂ monitoring station (71°N, Alaska)¹⁸ to investigate the response of summer CO₂ uptake to temperature variability in arctic and boreal ecosystems. The indicator we use is the summer CO₂ drawdown (SCD), calculated as the CO₂ concentrations difference between the first week of July and the last week of August in the detrended CO₂ record (see Methods, Supplementary Fig. 1). We show that there is a significant negative inter-annual correlation between SCD with summer land temperature north of 50° N over the last three decades, and such significant negative correlation only occurred during the period 1979–1995 rather than during the period 1996–2012. The causes for the observed temporal shift in the correlation are explored based on multiple satellite-derived products^{19, 20, 21}, an ensemble of terrestrial carbon cycle models and simulations with an atmospheric transport model²².*” into “*The aim of this study was to understand the effect of temperature on summer CO₂ uptake in northern ecosystems and its decadal variation. We used the long-term record of atmospheric CO₂ concentrations from the Barrow atmospheric CO₂ monitoring station (71°N, 157°W, Alaska)²⁵ to calculate the summer CO₂ drawdown (SCD), which we used as an indicator of CO₂ uptake. SCD was calculated as the difference in the CO₂ concentration between the first week of July and the last week of August in the detrended CO₂ record (see Methods, Supplementary Fig. 1). Here, with simultaneous*

use of multiple satellite-derived products²⁶⁻²⁸, an ensemble of terrestrial carbon cycle models and simulations with an atmospheric transport model²⁹ (see Methods), we investigate the inter-annual correlation of the linkage between SCD and summer and its decadal change during 1979–2012.”. (Line 63–72 on page 4)

[**Comment 7**] L 86 These changes in gamma results should be presented in Fig. 1 (or else in supplementary).

[**Response**] We have added the following figure to present gamma results in the supplementary.

Figure R9. The inter-annual sensitivity of summer CO₂ drawdown (SCD) to summer temperature (γ_{SCD-T}) for 1979–1995 (blue) and 1996–2012 (red) periods. The summer temperature is calculated as the average for July and August across ecosystems north of 50°N (a), and the spatial average weighted by the potential emission sensitivities from FLEXPART over the vegetated land area within the multi-year mean summer footprint (b). * and ** indicate that partial correlation coefficient is statistically significant at $P < 0.05$ and $P < 0.01$, respectively.

[Comment 8] L 104 *No arctic ocean in the footprint- impossible.*

[Response] We have updated Figure S4 that now includes the ocean region. In addition, we have changed “*Using the FLEXPART Lagrangian particle dispersion model²³, we found that the summer footprint area of Barrow station is mainly restricted to regions of Siberia and Alaska (Supplementary Fig. 2).*” to “*The area of the summer flux footprint of the Barrow station calculated using the FLEXPART Lagrangian particle dispersion model³¹ was mainly restricted to the regions of Siberia and Alaska and the Chukchi and Beaufort Seas (Supplementary Fig. 4). A similar footprint area was found by a simulation of the sensitivity of the Barrow CO₂ measurements during the last week of August to terrestrial carbon fluxes 20 days before the measurements using the adjoint code of the Laboratoire de Météorologie Dynamique (LMDZ) atmospheric transport model (see Methods; Supplementary Fig. 5a-c).*”. (Line 110–117 on page 6)

[Comment 9] L 116 *It sounds as though Wang et al. at least considered the phase shift in their analysis. However, the caption to Figure S3 still implies a fixed window from DOY 178 to 235. What if you let this window vary based on the data? These results are possibly more important. Did they shift the temperature window accordingly in their regression analysis?*

[Response] In the prior MS, we calculated SCD as the difference of CO₂ concentration between the climatological trough date (day of year 235) and climatological spring zero-crossing date (day of year 178), which can effectively

represent a period of strong CO₂ drawdown from the available CO₂ record. Since spring-zero crossing date and trough date could vary across years, we also calculate SCD using inter-annually varying spring zero-crossing date and trough date. In the partial-correlation analyses, the period used to calculate summer climatic variables (temperature, precipitation and radiation) is also inter-annually varying, and is consistent with that used to compute SCD. Our results are in line with that based on the fixed-window approach, suggesting that the varying phase of CO₂ seasonal cycle should not affect our main findings. The details can be seen in the response of *Comment 1* by Reviewer #1.

[**Comment 10**] *L 162 This assumes that temperature variance is not changing over time-not a valid assumption.*

[**Response**] Before performing correlation analyses, we calculate the standard deviation of daily temperature between July and August for each year and find that there is no significant trend in the standard variation of summer temperature during the entire period (1979–2012) (Figure R3 and R4), suggesting that the temperature variance is relatively stable through time. Please see response to *Comment 1* for more details.

[**Comment 11**] *L 177 this is interesting because increased soil moisture could promote CH₄ instead of CO₂ losses*

[**Response**] Thanks for your inspiring comment. It would be interesting to see how

increased soil moisture affects the responses of atmospheric methane to climate change in the future study.

[Comment 12] *L 183 I would restructure discussion and start with total ecosystem respiration (TER) and then move towards heterotrophic respiration (HR).*

[Response] Following your suggestion, we have changed “*First, we constructed a heterotrophic respiration (HR) field by subtracting satellite-derived NPP⁸ from net biome production (NBP) derived from the MACC (Monitoring Atmospheric Composition and Climate) atmospheric CO₂ inversion²⁸. Our results showed that the interannual partial correlation of summer HR with summer temperature (R_{HR-T}), whilst controlling for the effects of precipitation and cloudiness, was significant ($P < 0.01$) for the two periods, with coefficients of 0.75 ± 0.14 and 0.69 ± 0.06 for 1979–1995 and 1996–2012 (Supplementary Fig. 10a), respectively. In addition, similar results are also obtained with ecosystem respiration computed by subtracting flux-tower-based GPP from MACC NBP (Supplementary Fig. 10b). Second, we used simulated HR from the ensemble of nine terrestrial carbon cycle models that provide bottom-up estimates of carbon cycle processes for Global Carbon Project (see Methods).” into “*We tested the hypothesis that the increased response of respiration to temperature was responsible for the change in R_{SCD-T} . We inferred the ecosystem respiration (RECO) field by subtracting GPP based on flux-tower data²⁸ from net biome production (NBP) derived from the net CO₂ flux from the Monitoring Atmospheric Composition and Climate (MACC) atmospheric CO₂ inversion³⁹. The interannual partial correlation of summer**

RECO with summer temperature (R_{RECO-T}), while controlling for the effects of precipitation and cloudiness, was significant ($P < 0.01$) for both study periods, but the coefficients of 0.78 ± 0.13 for 1979–1995 and 0.72 ± 0.05 for 1996–2012 (Supplementary Fig. 18a) do not suggest a correlation becoming more positive in the later period. We also calculated heterotrophic respiration (HR) by subtracting satellite-derived NPP²⁷ from the MACC inversion net CO₂ flux and then calculated the correlations R_{HR-T} . We obtained results similar to those for R_{RECO-T} (Supplementary Fig. 18b).”. (Line 216–227 on page 11)

[Comment 13] *L 208 I suggest you go back and re-read the Keeling paper and discuss temperature sensitivity in a broader context*

[Response] Following your suggestion, we have added the following text into the revised MS.

“In addition, the overall decline in the stimulating effect of temperature on carbon uptake in spring and summer suggested that the previously reported warming-induced increase in the duration of carbon uptake and then the seasonal CO₂ amplitude³⁰ would become weak, consistent with a recent study demonstrating that the increase in seasonal CO₂ amplitude was due much more to CO₂ fertilization than to climate change².”. (Line 250–255 on page 12)

[Comment 14] *Assume that air sea gas exchange plays little role in changing C uptake over time. (Fig S2)*

[Response] Following your above-mentioned constructive comment, we have shown that changes in Arctic sea surface temperature are not mainly responsible for the emerging negative temperature impact on summer C uptake. The details can be seen in the response of *Comment 3* by Reviewer #1.

To Reviewer #2

[General Comment] *This is an interesting paper containing original analyses and that is potentially worthy of publication. The data, data analysis, and central conclusions are interesting, seem solid, and seem eventually publishable. However, the paper needs major rewriting for clarity, to accurately report the intent and results, for grammar, the literature, and for accuracy. Much more care needs to be given to insuring that each word and phrase portrays the situation accurately, clearly, and that what is meant is accurately portrayed. Non-careful writing has led to many inaccurate statements because the intent of the writing is not accurately or appropriately conveyed. I suggest that some of the fluent English language authors, or an editor, take a more active role in the re-writing and editing of the manuscript. Much more work is needed before this manuscript is ready for publication. Below I give some examples. This list is not exhaustive, rather it is meant to point out the type of problems that exist with the manuscript. If only the items below are fixed, and that the sense of this review is not conveyed to the entire manuscript, it will still not be ready for publication.*

[Response] We strongly agree with the reviewer's concern on our English writing.

Thank you for your great patience in listing the problems that exist with the manuscript.

Following your suggestion, we have requested several fluent English speakers to play a more active role in the re-writing and editing of the whole manuscript. We hope that the revised manuscript could satisfactorily addresses all of your concerns.

[Specific comments]

[Comment 1] L20-21. *As written, this is a ridiculous statement. It shows non-careful formulation, a lack of specificity, and possibly a lack of knowledge of the literature. Certainly, most of the previous high latitude carbon cycle have been done in the summer. The major lack is information on the fall, winter, and spring periods. See work done at NARL in the 60s and during IBP for example and early eddy covariance work in the Arctic. The authors may mean a particular type of inversion study, but they do not say that.*

[Response] We strongly agree with the reviewer's comment, and we have rephrased it as "*Most recent studies of the northern hemisphere carbon exchange based on seasonal observations of atmospheric CO₂ concentration have focused on spring^{1,2} and autumn³, but the impact of climate change on summer carbon fluxes remains unclear.*". (Line 20–22 on page 2)

[Comment 2] L27. *Why surprising? Change "surprising" to "however". L27-32. Awkward, confusing, unclear what is meant. Rewrite carefully and clearly.*

[Response] We have rephrased this sentence as: "*A refined analysis at the decadal scale reveals, however, strong differences between the earlier (1979–1995) and later (1996–2012) periods of the data set, with the significant negative correlation only in the later period. This temporal change in the correlation between summer CO₂ uptake and temperature is due to the disappearance of the positive response of summer vegetation productivity to temperature that prevailed in the earlier period. This emerging negative response of summer carbon uptake to temperature, together with the*

recently reported decline of the positive effect of warmer years on spring carbon uptake², highlights a diminished overall positive effect of warming on high-latitude carbon uptake, which could fundamentally alter the high-latitude terrestrial carbon balance.” (Line 28–37 on page 2)

[Comment 3] L33. *What type of disturbances? How does temperature cause Arctic disturbances?*

[Response] Prior studies have shown that warming leads to the intensification in both fire frequency and severity, as well as larger areas burned in arctic and boreal ecosystems (e.g. Kasischke and Turetsky, 2006; Flannigan et al., 2009; Hu et al., 2010). To be more specific, we have changed “disturbances” into “*fire disturbances*”.

[Comment 4] L33-34. *This is debated. There are many reports of many northern areas switching from a C sink to an annual C source in recent decades. Review the literature starting in the 1990s (see e.g. Oechel et al. Nature 1993, 2000 and 2014 to cite a few).*

[Response] Following your suggestion, to be more accurate, we have changed “*act as an important atmospheric CO₂ sink in the contemporary global carbon cycle*” into “*Arctic and boreal ecosystems play an important role in the global carbon cycle, and their carbon cycle responses to climate change become a major global concern^{4,5}.”*

(Line 38–39 on page 3)

[Comment 5] L36, 39 *Here and elsewhere. Mostly, what is discussed is C fluxes not C*

cycles. Use C fluxes.

[Response] We have used carbon fluxes instead of carbon cycles in all places except the first sentence of the introduction.

[Comment 6] L38. *This is not true. You are cherry picking the results and papers cited.*

L39. *This is simply not true. The greatest amount of attention has been given to summer.*

This is a ridiculous assertion. If you mean a particular type of study, then say so.

[Response] We strongly agree with the reviewer's comment. Following your suggestion, we have changed "*Most of prior studies are concentrated on the edges of growing season (spring and autumn)^{1, 2, 3, 4}, but little attention has been given to summer.*" into "*Numerous field studies in high-latitude ecosystems have been conducted in the summer¹⁰⁻¹⁸, but the results of these local experiments are limited in extent and are difficult to scale up to larger regions. Observations of atmospheric CO₂ concentration from atmospheric stations at high latitudes provide complementary monitoring of the dynamics of carbon exchange in northern ecosystems. But most of these studies have concentrated on the amplitude of the seasonal cycle or the spring and autumn boundaries of the growing season¹⁻³, little attention has been given to the summer.*". (Line 42–48 on page 3)

[Comment 7] L42-44. *Not true. The spring has ideal climatic conditions for C uptake.*

It is just that the plants have not developed yet. Phenology may be controlling spring/early summer uptake.

[Response] Thanks for the reviewer bringing up this point. Following your suggestion, we have removed “unlike the edges of growing season” in the revised manuscript. Specifically, we have changed “*Knowledge of the response of summer carbon cycle to temperature is important for annual carbon sequestration, given that the summer has the highest vegetation productivity and strongly contributes to year-to-year variations in high-latitude land carbon uptake¹³. Furthermore, unlike the edges of growing season, the summer has generally the most favourable climate conditions that could maximize the CO₂ uptake capacity.*” into “*Summer has the highest vegetation productivity, strongly contributes to interannual variations in terrestrial carbon uptake¹⁹ and generally has the most favorable climatic conditions for CO₂ uptake.*”.

(Line 49–51 on page 3)

[Comment 8] L 48. Unclear what is meant by the "response of the carbon cycle". What is "the" carbon cycle?

[Response] Sorry for our awkward English expression. We have removed “the” in this sentence. Moreover, we have asked several fluent English speakers to improve the English writing.

[Comment 9] L50. Simply not true. The summer carbon cycle is the best studied of all seasons. See results from the Arctic IBP from the early 1970s and following sutides.

[Response] Following your suggestion, we have cited the early pioneer work on the summer carbon cycle and included the following sentence “*Numerous field studies in*

high-latitude ecosystems have been conducted in the summer¹⁰⁻¹⁸” in the revised MS.

(Line 42–43 on page 3)

[Comment 10] *L93. What does "Main" summer footprint mean? 50% of the entire footprint? Be specific.*

[Response] We thank the reviewer for bringing up this point. In this study, the summer footprint map, which is indicated by potential emission sensitivity from FLEXPART (or surface flux sensitivity from LMDZ), is used to weight climate variables (e.g. temperature). We then analyze the relationship between summer carbon uptake and the spatial average of weighted temperature. We did not set a cutoff value of sensitivities to select the summer footprint for the weighting, because a large fraction of the land surface had low sensitivity values. For example, values of potential emission sensitivity from FLEXPART above 0.1 s constituted 93.35% of the land-surface signal, and values above 1 s constituted 41.70% of the land-surface signal.

In order to well resolve the reviewer’s concern, we have changed “**a, b**, *Time series of anomaly of summer CO₂ drawdown (SCD, black line) and summer temperature (T, red line) calculated as the average temperature for July and August across ecosystems north of 50°N (a), and over the main summer footprint region of Barrow CO₂ station (b).*” into “**a, b**, *Time series of anomalies of summer CO₂ drawdown (SCD, black line) and summer temperature (T, red line) calculated as the average for July and August across ecosystems north of 50°N (a), and the spatial average weighted by the potential*

emission sensitivities from FLEXPART over the vegetated land area within the multi-year mean summer footprint (b).” (Line 289–293 on page 15)

Furthermore, we have changed “*Summer CO₂ concentrations observed at BRW were mainly from terrestrial carbon fluxes within the summer footprint area, so we also calculated the spatial averages of the climatic variables during the earlier and later period by weighting potential emission sensitivity from FLEXPART over the vegetated land area within the multiple-year mean summer footprint area.*” into “*All climatic variables for the main analysis were calculated as spatial averages over the vegetated land north of 50°N. Vegetated land was defined as areas with a mean annual NDVI > 0.1 for 1982–2012. We also calculated the spatial averages of the climatic variables, as part of the robustness tests, by weighting the calculated sensitivities (potential emission sensitivity from FLEXPART and surface flux sensitivity from LMDZ) of different vegetated land areas within the multiple-year mean summer-footprint area. We did not set a cutoff value of sensitivities to select the summer footprint for the weighting, because a large fraction of the land surface had low sensitivity values. For example, values of potential emission sensitivity from FLEXPART above 0.1 s constituted 93.35% of the land-surface signal, and values above 1 s constituted 41.70% of the land-surface signal.*”. (Line 445–454 on page 23)

[Comment 11] L109. "does not affect" should be "does affect"

[Response] In this study, we documented that a significant negative effect of

temperature on summer CO₂ uptake has recently emerged, based upon the analysis relating summer carbon uptake to summer land temperature north of 50°N. Since terrestrial carbon fluxes impacting the Barrow CO₂ concentration are not spatially uniform in the high-latitudes, we performed the additional analysis using temperature spatially weighted with the footprint intensity derived from FLEXPART. This analysis confirmed our main finding. We thus dare to argue that our previous expression is not mistaken.

To be more clear, we have changed “*When interpreting this shift in R_{SCD-T} , one could argue that the contribution of terrestrial carbon fluxes to the Barrow CO₂ concentration is not uniform over all the high latitudes. Using the FLEXPART Lagrangian particle dispersion model²³, we found that the summer footprint area of Barrow station is mainly restricted to regions of Siberia and Alaska (Supplementary Fig. 2). We therefore calculated R_{SCD-T} using climate variables averaged over this footprint area (see Methods), and found that R_{SCD-T} shifted from a non-significant positive value ($R = 0.01$, $P = 0.97$) to a significant negative value ($R = -0.61$, $P < 0.05$) (Fig. 1b), suggesting that the choice of spatial domain for averaging climate variables does not affect our finding.*” into “*We first calculated R_{SCD-T} using climatic variables spatially weighted with the footprint intensity over the flux footprint area of the CO₂ record of the Barrow station (see Methods), because terrestrial carbon fluxes impacting the Barrow CO₂ concentration are not spatially uniform in the high-latitudes. The area of the summer flux footprint of the Barrow station calculated*

using the FLEXPART Lagrangian particle dispersion model³¹ was mainly restricted to the regions of Siberia and Alaska and the Chukchi and Beaufort Seas (Supplementary Fig. 4). A similar footprint area was found by a simulation of the sensitivity of the Barrow CO₂ measurements during the last week of August to terrestrial carbon fluxes 20 days before the measurements using the adjoint code of the Laboratoire de Météorologie Dynamique (LMDZ) atmospheric transport model (see Methods; Supplementary Fig. 5a-c). We also performed 40- and 60-day back-trajectory calculations using LMDZ and found that the footprint area for fluxes influencing the SCD did not change significantly compared to a 20-days influence (Supplementary Fig. 5). $R_{\text{SCD-T}}$ between the two study periods still decreased when calculated using temperature spatially weighted with the footprint intensity. $R_{\text{SCD-T}}$ shifted from a non-significant positive value ($R = 0.01$, $P = 0.97$) to a significant negative value ($R = -0.61$, $P < 0.05$) in this test (Fig. 1b). $R_{\text{SCD-T}}$ also shifted when the footprint was derived from 40- and 60-day back-trajectory calculations rather than with a value of 20 days (Supplementary Fig. 6).” (Line 107–124 on page 6)

[Comment 12] L202. "profoundly". No profound alteration has been demonstrated in this paper. Remove "profoundly".

[Response] We have removed “profoundly” in the revised MS.

[Comment 13] L203. Shifting to a significantly "negative state" is unclear. Rewrite the sentence.

[Response] We have changed “*Our findings provide evidence that the effect of temperature on summer CO₂ uptake in arctic and boreal ecosystems has been altered at the interannual timescale in the last three decades, shifting toward a significantly negative state, most likely due to the lowered effect of temperature on summer productivity (Fig. 3).*” into “*We demonstrated that the effect of temperature on summer CO₂ uptake in arctic and boreal ecosystems has been altered in the last three decades. A significant negative effect of temperature on summer CO₂ uptake has recently emerged, that is, warmer years coincide with less summer uptake, which is most likely due to the reduced effect of temperature on summer productivity (Fig. 3).*”. (Line 237–241 on page 12)

[Comment 14] L206 Change to: to "increased" net carbon..."

[Response] Done.

[Comment 15] L210-213. Cite papers going back to IBP (1970s) and since including those in nature.

[Response] Following your suggestion, we have included the early pioneer work (e.g. Oechel et al., 1995; Vourlitis and Oechel, 1999; Oechel et al., 2000; McFadden et al., 2003; Lund et al., 2010; Natali et al., 2010) in the revised manuscript.

[Comment 16] Fig 3. Figure hard to interpret. Add variable "+" and "-" to indicate varying positive and negative effects.

[Response] Following your suggestion, we have added “+” and “-” to indicate the positive and negative effect, respectively.

[Comment 17] L243. *Performed "CO₂ 'measurement' analysis" is unclear. What is measurement analysis? Explain.*

[Response] Sorry for our confusion expression. We have changed this sentence into: “*D.L. performed all statistical analyses based on the Barrow CO₂ observations and gridded carbon fluxes*”. (Line 280–281 on page 14)

[Comment 18] L306-309. *What tower observations? How was this done? Where are the results?*

[Response] Sorry for this confusion. In this study, we did not directly use eddy-covariance measurements of CO₂ flux at the station (or tower) level, but used the global gross primary productivity (GPP) product (0.5° × 0.5°, monthly, 1982–2011) that is generated by integrating a global network of eddy covariance sites, satellite remote sensing, and meteorological data in a machine learning algorithm. This global product has been widely used to understand the spatio-temporal dynamics of the global carbon fluxes, and to benchmark process-based land models.

To be clearer, we have changed “*The second one is the upscaled gross primary productivity (GPP) from flux tower observations to the global scale (0.5° × 0.5°, monthly, 1982–2011) using the machine learning technique based on remote sensing*”

*indices and climate data*²¹.” into “*The second was the global gross primary productivity (GPP) product (0.5° × 0.5°, monthly, 1982–2011) generated by integrating a global network of eddy covariance sites, satellite remote sensing and meteorological data in a machine-learning algorithm*²⁸. *This global product has been widely adopted to aid the understanding of the spatiotemporal dynamics of the global carbon cycle*⁵¹ *and to benchmark process-based terrestrial models*^{28,52,53}.” (Line 391–396 on page 21)

[Comment 19] L354-355. *What does this mean? How do you know this? How do you know that summer CO₂ conc. Observed at BRW are mainly from terrestrial sources within the summer footprint area. Reference the SI and explain. How did you rule out ocean sources or sinks?*

[Response] We agree with the reviewer’s concern that Point Barrow is located on the edge of the Arctic Ocean and its CO₂ variation could be influenced by ocean carbon fluxes according to our updated summer footprint area (Figure R6). To be more accurate, we have changed “*Summer CO₂ concentrations observed at BRW were mainly from terrestrial carbon fluxes within the summer footprint area, so we also calculated the spatial averages of the climatic variables during the earlier and later period by weighting potential emission sensitivity from FLEXPART over the vegetated land area within the multiple-year mean summer footprint area.*” to “*All climatic variables for the main analysis were calculated as spatial averages over the vegetated land north of 50°N. Vegetated land was defined as areas with a mean annual NDVI > 0.1 for 1982–2012. We also calculated the spatial averages of the climatic variables, as*

part of the robustness tests, by weighting the calculated sensitivities (potential emission sensitivity from FLEXPART and surface flux sensitivity from LMDZ) of different vegetated land areas within the multiple-year mean summer-footprint area.” (Line 445–450 on page 23)

In addition, in order to assess the potential impact of changes in air-sea gas exchanges on our main finding, we perform the partial-correlation analysis between summer carbon uptake and land temperature after statistically controlling for the effect of Arctic sea surface temperature (SSTs). The following text has therefore been added into the revised MS.

“We further investigated whether the shift in R_{SCD-T} may have been indirectly related to changes in the onset date of the Arctic sea-ice melt. The date of the onset of melting of sea ice advanced at rates around 2.8 day decade⁻¹ during 1979–2004 at Chukchi and Beaufort Seas³². Earlier melting could increase the air-sea CO₂ flux in the summer and potentially contribute to the variations of summer CO₂ concentration at Barrow. We would then expect a correlation between SCD and Arctic sea-surface temperature (SST), because SST is closely correlated with the percentage of open water in the Arctic during the summer³³. We tested this hypothesis by calculating the correlation between SCD and summer (July-August) SSTs north of 50°N for the two study periods but found no significant correlation for either period (Supplementary Fig. 13). A possible indirect effect of SSTs on the shift in R_{SCD-T} was investigated using the partial-correlation between SCD and land temperature after controlling for the

effects of cloudiness, precipitation and Arctic SSTs. R_{SCD-T} shifted similarly to the original calculation (Supplementary Fig. 14), suggesting that the impact of changes in air-sea exchanges on the SCD changes was limited.”. (Line 155–168 on page 8)

To Reviewer #3

[General Comments]

The manuscript demonstrates a significant change in the interannual relationship between summer land temperature and summer drawdown of atmospheric CO₂ at high northern latitudes over recent decades. Careful comparison of the changing correlations between temperature and CO₂ drawdown with proxy data for photosynthesis and ecosystem respiration indicate that the change in temperature response most likely results from heat and drought stress on plant productivity as a result of an increase in very warm weather. The data are from well-established measurement programs, the analysis is straightforward, the implications are important, and the paper is very well-written. I recommend publication with minor revisions as explained below.

[Response] Thanks for your encouraging comments. We revised our MS carefully following your comments and suggestions.

[Comment 1] *The authors use a backward-in-time Lagrangian particle transport model (FLEXPART) to calculate upstream "concentration footprints" for the CO₂ measurements at Point Barrow. The summertime footprint is used to define averaging areas for the various proxy data (NDVI, NPP, etc) that are used to investigate hypothetical mechanisms that might explain the change in temperature response of CO₂ drawdown. This is a reasonable method and is based on well-established meteorological data and models, but it's surprising that the resulting footprint was*

localized to Alaska and eastern Siberia.

[Response] We have updated Figure S2 that now includes the ocean region. In addition, we have changed “*Using the FLEXPART Lagrangian particle dispersion model²³, we found that the summer footprint area of Barrow station is mainly restricted to regions of Siberia and Alaska (Supplementary Fig. 2).*” to: “*The area of the summer flux footprint of the Barrow station calculated using the FLEXPART Lagrangian particle dispersion model³¹ was mainly restricted to the regions of Siberia and Alaska and the Chukchi and Beaufort Seas (Supplementary Fig. 4). A similar footprint area was found by a simulation of the sensitivity of the Barrow CO₂ measurements during the last week of August to terrestrial carbon fluxes 20 days before the measurements using the adjoint code of the Laboratoire de Météorologie Dynamique (LMDZ) atmospheric transport model (see Methods; Supplementary Fig. 5a-c).*”. (Line 110–117 on page 6)

Furthermore, the summer main footprint areas are further confirmed based on the adjoint code of the LMDZ atmospheric transport model (Figure R10). According to our updated summer footprint area, Point Barrow is located on the edge of the Arctic Ocean and its CO₂ variation could be influenced by ocean carbon fluxes. We have also performed additional analyses to rule out the potential impact of changes in air-sea gas on our main finding (see details in **Comment 3** by the Reviewer #1).

[Comment 2] *The authors cite studies by Gray et al (2014) and Zeng et al (2014) both*

of which attribute changes in high-latitude CO₂ seasonality to increases in agricultural production in mid-latitudes. A recent analysis by Barnes et al (2016) showed that CO₂ seasonality at Point Barrow is strongly influenced by ecosystem processes in temperate latitudes, and that transport by baroclinic waves carries this seasonality into the high Arctic. I suspect that the overall conclusions of the present manuscript do not depend sensitively on the footprint over which the proxy data are averaged. It seems likely that temperate ecosystems may also show changes in temperature response with recent warming. But the authors should address the issue of long-distance transport of seasonal CO₂ variations from lower latitudes. How sensitive are their conclusions to the particular footprint calculations they used? Perhaps 20 days of back-trajectory calculations are insufficient to track CO₂ backward from Point Barrow to temperate croplands, pastures, and forests? At the very least the manuscript should cite recent work on long-distance meridional transport of CO₂ seasonality and consider the implications for their results.

[Response] To address whether 20 days of back-trajectory calculations are sufficient to track CO₂ backward from Barrow to temperate ecosystems, we have also performed 40 and 60 days back-trajectory calculations starting from the last week of August based on the adjoint code of the LMDZ atmospheric transport model. As shown in the Figure R10, the main summer footprint area of Barrow station, based on the three different days of back-trajectory calculations, is restricted to regions of eastern Siberia, Alaska and its surrounding seas. We then calculated R_{SCD-T} using climate variables averaged over the three different summer footprint areas, and found that R_{SCD-T} shifted from a

non-significant positive value to a significant negative value (Figure R11), suggesting that the choice of particular number of days in back-trajectory calculation does not affect our main finding.

The analysis made by Barnes et al (2016) showed that CO₂ signals and seasonality can be transported from mid-latitude to high-latitude along surfaces of constant potential temperatures. They showed that the seasonal CO₂ amplitude at ~70°N in most of the troposphere is more sensitive to seasonality of surface fluxes at ~30°N than at ~60°N. While, a close examination of their Figure 11c indicated that the near-surface seasonal CO₂ amplitude (below 850hPa in their Figure 11c) is more sensitive to the seasonality of surface fluxes at 70°N than from other latitudes (30°N -60°N). Their Figure 12 also showed that the seasonal amplitude of surface measurements along the isentrope of 265K is more sensitive to fluxes at 70°N than at other latitudes. Since the Barrow (71.3°N, 156.6°W) is a surface station (11 meters above sea level), with barometric pressure around 1000 hPa and air temperature near -20°C in March (consistent with Figure 12 in Barnes et al., 2016), our simulation showing that the main footprint area are Siberia and Alaska therefore does not differ from the results of Barnes et al. (2016).

Figure R10. Summer footprint area for Barrow derived from LMDZ model. The left (a-c), middle (d-f) and right (g-i) panel are footprint tracing 20, 40 and 60 days back from the last week of August, respectively.

Figure R11. The time series of detrended anomaly of summer CO₂ drawdown (SCD, black line) and summer temperature (T, red line) calculated as the average temperature for July and August over the main summer footprint region of Barrow CO₂ station. The summer footprint is calculated using LMDZ model with 20 days (a), 40 days (b) and 60 days (c) back-trajectory since the last week of August.

In order to resolve the reviewer’s concern, we have changed “Using the FLEXPART

Lagrangian particle dispersion model²³, we found that the summer footprint area of Barrow station is mainly restricted to regions of Siberia and Alaska (Supplementary Fig. 2). We therefore calculated R_{SCD-T} using climate variables averaged over this footprint area (see Methods), and found that R_{SCD-T} shifted from a non-significant positive value ($R = 0.01$, $P = 0.97$) to a significant negative value ($R = -0.61$, $P < 0.05$) (Fig. 1b), suggesting that the choice of spatial domain for averaging climate variables does not affect our finding.” into *“The area of the summer flux footprint of the Barrow station calculated using the FLEXPART Lagrangian particle dispersion model³¹ was mainly restricted to the regions of Siberia and Alaska and the Chukchi and Beaufort Seas (Supplementary Fig. 4). A similar footprint area was found by a simulation of the sensitivity of the Barrow CO₂ measurements during the last week of August to terrestrial carbon fluxes 20 days before the measurements using the adjoint code of the Laboratoire de Météorologie Dynamique (LMDZ) atmospheric transport model (see Methods; Supplementary Fig. 5a-c). We also performed 40- and 60-day back-trajectory calculations using LMDZ and found that the footprint area for fluxes influencing the SCD did not change significantly compared to a 20-days influence (Supplementary Fig. 5). R_{SCD-T} between the two study periods still decreased when calculated using temperature spatially weighted with the footprint intensity. R_{SCD-T} shifted from a non-significant positive value ($R = 0.01$, $P = 0.97$) to a significant negative value ($R = -0.61$, $P < 0.05$) in this test (Fig. 1b). R_{SCD-T} also shifted when the footprint was derived from 40- and 60-day back-trajectory calculations rather than with a value of 20 days (Supplementary Fig. 6).”*. (Line 110–124 on page 6)

In addition, we have also added the following paragraph into the method section

“Summer CO₂ uptake from the Barrow CO₂ data”.

“The second method used to define the summer footprint area was based on the adjoint code of the LMDZ atmospheric transport model³⁹. The adjoint code calculates the partial derivatives of concentration measurements for fluxes at any time before a given CO₂ observation. We calculated the partial derivatives for the mean CO₂ measurements made at Barrow during the last week of August for fluxes at daily resolutions since the start of July, for all years of this study. Integrating the derivatives for a given number of days (Supplementary Fig. 5) within one year quantified the change in CO₂ concentration at Barrow per unit change in flux (kg C m⁻² h⁻¹) for all days before the measurement. We then averaged the derivatives for each year between 1979 and 2012 to obtain a multi-year mean area of the summer footprint.”. (Line 363–372 on page 19–20)

References

- Flannigan, M. D. *et al.* Impact of climate change on fire activity and fire management in the circumboreal forest. *Glob. Chang. Biol.* **15**, 549–560 (2009).
- Galbraith, P. S. & Larouche, P. Sea-surface temperature in Hudson Bay and Hudson Strait in relation to air temperature and ice cover breakup, 1985-2009. *J. Mar. Syst.* **87**, 66–78 (2011).
- Hu, F. *et al.* Tundra burning in Alaska: linkages to climatic change and sea ice retreat.

- J. Geophys. Res.* **115**, G04002, doi:10.1029/2009JG001270 (2010).
- Kasischke, E. S., Rupp, T. S. & Verbyla, D. L. Fire trends in the Alaskan boreal forest. In Alaska's changing boreal forest. Edited by F.S. Chapin III, M.W. Oswood, K. Van Cleve, L.A. Viereck, and D.L. Verbyla. Oxford University Press, New York. pp. 285–301 (2006).
- Lafleur, P. M. & Humphreys, E. R. Spring warming and carbon dioxide exchange over low Arctic tundra in central Canada. *Global Change Biology* **14**, 740–756 (2007).
- Lund, M. *et al.* Variability in exchange of CO₂ across 12 northern peatland and tundra sites. *Global Change Biology* **16**, 2436–2448 (2010).
- Markus, T., Stroeve, J. C. & Miller, J. Recent changes in Arctic sea ice melt onset, freezeup, and melt season length. *J. Geophys. Res.* **144**, C12024 (2009).
- McFadden, J. P. *et al.* A regional study of the controls on water vapor and CO₂ exchange in arctic tundra. *Ecology* **84**, 2762–2776 (2003).
- Natali, S. M. *et al.* Effects of experimental warming of air, soil and permafrost on carbon balance in Alaskan tundra. *Global Change Biology* **17**, 1394–1407 (2011).
- Oechel, W. C. *et al.* Recent change of arctic tundra ecosystems from a net carbon dioxide sink to a source. *Nature* **361**, 520–523 (1993).
- Oechel, W. C. *et al.* Acclimation of ecosystem CO₂ exchange in the Alaskan Arctic in response to decadal climate warming. *Nature* **406**, 978–981(2000).
- Vourlitis, G. L. & Oechel, W. C. Eddy covariance measurements of net CO₂ flux and energy balance of an Alaskan moist-tussocktundra ecosystem. *Ecology* **80**, 686–

701 (1999).

Reviewers' comments:

Reviewer #1 (Remarks to the Author):

Dear Authors,

The authors have performed considerably more analyses to indicate that their results are robust. In particular, considering the effects of a temporally varying summer window of analysis and the effects of sea-ice. In considering these other factors the authors have concluded that their conclusions remain unaltered. However, in some instances it appears that using a temporally varying window lead to a change in the sign of correlation of temperature and SCD. In the instance of sea-ice it seems as though their correlations were slightly enhanced even though they looked at sea surface temperatures and not sea ice extent suggesting perhaps an interaction between sea surface and land surface temperatures. It should also be noted that the solubility of CO₂ is decreased with increases in sea-surface temperature. Lastly, their analysis of diurnal temperature over time suggests that variance has not changed, but it was my understanding that minimum temperatures are increasing faster than maximum temperatures (see early work by Easterling) leading to a decrease in the seasonal range of temperatures. So the variance within any year should be going down at the global scale, but I am not sure if this is true in the Boreal/Arctic zone. Overall, this is a nice analysis.

Reviewer #2 (Remarks to the Author):

I appreciate the work in revising this manuscript for Nature Communications. Many improvements have been made and are appreciated. The authors have done a better job citing previous work and being more careful in the phrasing and claims made.

The main point of this paper is stated by the title: "Emerging negative impact of warming on summer carbon uptake in northern ecosystems." However, the main data used in justifying the conclusions of the paper is atmospheric concentration data. The authors do not show the change, through time, in net summer uptake, and therefore cannot draw the conclusions that they put forward. For example, the authors conflate concentration and CO₂ uptake: "summer CO₂ uptake (the difference in CO₂ concentration between the first week of July and the last week of August)" (lines 25-27). These are simply two different parameters and cannot be used interchangeably. Fluxes and concentrations can be affected by very different processes and source areas. Summer uptake, for the study region is never shown yet it is the basis of the paper. For this paper to move to be publishable, the authors need to calculate, validate, and present actual CO₂ fluxes. To do that, they need to convincingly show that their calculation of net CO₂ fluxes have a high degree of statistical confidence. They have not done that and this paper is therefore not appropriate for Nature Communications at this time. The authors attempt to justify the use of concentration and a surrogate for high latitude fluxes. However, if they want to make the claim that net summer uptake is becoming less sensitive to increasing summer temperatures, they need to work directly with fluxes, not concentrations.

One approach might be to refocus the paper on fluxes derived from satellite and model output. This would allow the authors to present and defend the calculated fluxes over the domain and discuss possible controls. The concentration data could be added for support or comparison, but not as key evidence in the paper.

Specific comments and observations:

L 31-34. Summer uptake is not shown. Net uptake is the result of ecosystem respiration and photosynthesis. Respiration is only known from model outputs and ecosystem carbon models for the Arctic are known to vary dramatically. Averaging a number of disparate model outputs does

not necessarily give the correct resultant flux.

L 63-68. Summer CO₂ drawdown maybe be an “indicator” of CO₂ uptake. However, the authors have not made the definitive quantitative link between SCD and net CO₂ uptake. This is a critical step missing from this analysis and presentation.

L 126-128. Randomly selecting 14 of 16 years of data seems pointless since almost all of the data is being used in each analysis. The full data sets should be used.

L 174. As stated in L 180, NDVI is only a proxy for productivity. They are not synonymous.

L 186-189. These are both indirect methods. The satellite derived NPP model has large uncertainties. The GPP data is poorly constrained by data.

L193-200. The number of hours when canopy photosynthesis is reduced by high canopy temperature has not been shown. Even on warm days, much of the day is below the temperature optimum for photosynthesis. Water stress might contribute to reduced photosynthesis. This is speculative as not data is shown.

L 209-212. Soil moisture and drought are two separate things. High temperature can cause drought even with high soil moisture. This is likely to occur, e.g., when root resistance is high due to low soil temperatures as occurs in Arctic soils.

L228-229. These models perform poorly against data. The mean or ensemble average of a number of disparate model outputs, one does not necessarily yield a true estimate of HR.

L231-234. If respiration continues to be depending on temperature, and if temperature continues to rise, it is unclear why this might not cause the purported decrease in summer NEE.

L236-238. Discussion. The statement that “We demonstrated that the effect of temperature on summer CO₂ uptake in arctic and boreal ecosystems has been altered in the last three decades”. There is no presentation of the actual rates of CO₂ uptake over the last three decades. There is inference, but not verified data. Therefore, this statement cannot be made. Only statements about the change in rates of drawdown with summer temperature can be made. The paper overstates the conclusions possible.

L257. The statement that “High-latitude ecosystems have not shifted from a carbon sink to a carbon source” cannot be made base on this paper or on the one paper cited that uses inference. There is considerable literature and uncertainty in this area, and it is insufficient to cite a single paper, especially one that is not definitive.

Reviewer #3 (Remarks to the Author):

The revised manuscript addresses the concerns regarding transport from lower latitudes which I had with the original through some additional simulations using longer integrations of the Lagrangian parcel model. In addition, the authors have added some important context in the recent literature which were missing in the original manuscript.

In addition to my own concerns, the authors have done a nice job in addressing the concerns of the other reviewer, particularly with regard to aliasing of phase and amplitude effects in the timeseries.

I find that the revised manuscript is an important and timely contribution to the emerging

understanding of carbon-climate feedbacks in high-latitude ecosystems, and recommend that it be published.

Reviewer #1

[General Comment]: *The authors have performed considerably more analyses to indicate that their results are robust. In particular, considering the effects of a temporally varying summer window of analysis and the effects of sea-ice. In considering these other factors, the authors have concluded that their conclusions remain unaltered. However, in some instances it appears that using a temporally varying window lead to a change in the sign of correlation of temperature and SCD. In the instance of sea-ice it seems as though their correlations were slightly enhanced even though they looked at sea surface temperatures and not sea ice extent suggesting perhaps an interaction between sea surface and land surface temperatures. It should also be noted that the solubility of CO₂ is decreased with increases in sea-surface temperature. Lastly, their analysis of diurnal temperature over time suggests that variance has not changed, but it was my understanding that minimum temperatures are increasing faster than maximum temperatures (see early work by Easterling) leading to a decrease in the seasonal range of temperatures. So the variance within any year should be going down at the global scale, but I am not sure if this is true in the Boreal/Arctic zone. Overall, this is a nice analysis.*

[Response] As stated by the reviewer, the sign of the correlation between summer CO₂ drawdown (SCD) and summer temperature during the earlier period becomes positive instead of staying negative if an interannually varying window was considered. But this positive value is not statistically significant at $P < 0.05$, and would then not overturn our main finding that the significant negative correlation between SCD and temperature emerged in the later

period. To be more accurate, we have rephrased the sentence “*We again found that R_{SCD-T} shifted, from 0.42 ($P = 0.12$) during 1979–1995 to -0.52 ($P < 0.01$) during 1996–2012 in that new test (Supplementary Fig. 12). From this, one can conclude that the detected shift in R_{SCD-T} is not an artifact of the method used to define SCD.*” as “*In this additional analysis, we found that R_{SCD-T} shifted from 0.42 ($P = 0.12$) during 1979–1995 to -0.52 ($P < 0.01$) during 1996–2012 (Supplementary Fig. 12). During the earlier period, R_{SCD-T} becomes positive instead of staying negative as it does when using the interannually varying window to define SCD. But the significant negative correlation between SCD and temperature still emerged in the later period leading to the main conclusion that this result is not affected by the method used to define SCD.*” (Line 153–158 on page 8)

In the second place, we strongly agree with the reviewer’s point that Arctic sea surface temperature (SST) should not be equivalent to Arctic sea-ice extent (SIE), although which could be strongly correlated. We therefore performed partial correlation analysis between SCD and summer air temperature (T) (R_{SCD-T}) whilst controlling for the effects of summer sea-ice extent (Figure R1). R_{SCD-T} shifted from no-significant negative value ($R = -0.16$, $P = 0.58$) during the earlier period (1979–1995) to significantly negative one ($R = -0.57$, $P < 0.05$) during the latter period (1996–2012). Our result emphasized that changes in summer Arctic sea-ice extent or Arctic sea-ice melt was not responsible for the shifted relationship between SCD and summer temperature.

Figure R1. Same as Figure 1, but calculated partial correlation between SCD and air temperature with controlling precipitation, cloudiness and Arctic sea-ice extent (SIE) during summer (July and August). The lines are time series of anomaly of summer CO₂ drawdown (SCD, black) and temperature (red) calculated as the average for July and August north of 50°N. The inset illustrates the interannual partial correlation coefficient between SCD and SIE. * and ** indicates that partial correlation coefficient is statistically significant at $P < 0.05$ and $P < 0.01$, respectively.

In order to resolve the reviewer’s concern, we have then replaced “*We would then expect a correlation between SCD and Arctic sea-surface temperature (SST), because SST is closely correlated with the percentage of open water in the Arctic during the summer³³. We tested this hypothesis by calculating the correlation between SCD and summer (July-August) SSTs north of 50°N for the two study periods but found no significant correlation for either period (Supplementary Fig. 13). A possible indirect effect of SSTs on the shift in R_{SCD-T} was investigated using the partial-correlation between SCD and land temperature after controlling for the effects of cloudiness, precipitation and Arctic SSTs. R_{SCD-T} shifted similarly*

to the original calculation (Supplementary Fig. 14), suggesting that the impact of changes in air-sea exchanges on the SCD changes was limited.” with “A possible effect of Arctic sea-ice melt on the shift in R_{SCD-T} was then investigated using the partial-correlation between SCD and land temperature after controlling for the effects of cloudiness, precipitation and summer Arctic sea-ice extent (SIE). R_{SCD-T} shifted similarly to the original calculation (Supplementary Fig. 13), suggesting that the impact of earlier Arctic sea-ice melt on the shift in R_{SCD-T} was limited.” (Line 164–169 on page 8–9)

Lastly, to address your concern that asymmetric warming between minimum and maximum temperature could reduce the diurnal temperature range (DTR) and then influence the temperature variance, we calculated changes in DTR over the last 60 years for the arctic and boreal region (north of 50°N). The decreasing DTR mainly occurs in 1950s and 1970s. For the period 1979–2012, the summer DTR over the arctic and boreal region did not show any significant change. Therefore, the temperature variance is relatively stable in the boreal and arctic zone over the last thirty years.

Figure R2. Changes in summer diurnal temperature range (DTR). The time series is calculated as the average for the region north of 50°N during summer (July to August). The thick black line denotes the 5-year smoothed DTR.

Reviewer #2

[Comment 1] *I appreciate the work in revising this manuscript for Nature Communications.*

Many improvements have been made and are appreciated. The authors have done a better job citing previous work and being more careful in the phrasing and claims made.

The main point of this paper is stated by the title: “Emerging negative impact of warming on summer carbon uptake in northern ecosystems.” However, the main data used in justifying the conclusions of the paper is atmospheric concentration data. The authors do not show the change, through time, in net summer uptake, and therefore cannot draw the conclusions that they put forward. For example, the authors conflate concentration and CO₂ uptake: “summer CO₂ uptake (the difference in CO₂ concentration between the first week of July and the last week of August)” (lines 25-27). These are simply two different parameters and cannot be used interchangeably. Fluxes and concentrations can be affected by very different processes and source areas. Summer uptake, for the study region is never shown yet it is the basis of the paper. For this paper to move to be publishable, the authors need to calculate, validate, and present actual CO₂ fluxes. To do that, they need to convincingly show that their calculation of net CO₂ fluxes have a high degree of statistical confidence. They have not done that and this paper is therefore not appropriate for Nature Communications at this time. The authors attempt to justify the use of concentration and a surrogate for high latitude fluxes. However, if they want to make the claim that net summer uptake is becoming less sensitive to increasing summer temperatures, they need to work directly with fluxes, not concentrations.

One approach might be to refocus the paper on fluxes derived from satellite and model output. This would allow the authors to present and defend the calculated fluxes over the domain and discuss possible controls. The concentration data could be added for support or comparison, but not as key evidence in the paper.

[Response] We agree with the reviewer that the use of actual net CO₂ fluxes in justifying our conclusion is straightforward. There are CO₂ flux products that might potentially be used in our analysis. Since large-scale direct observations of net ecosystem carbon exchanges are not available, the net CO₂ fluxes with spatially and temporally explicit maps are generally derived using the following three approaches: (1) the upscaling net CO₂ fluxes from a global network of eddy-covariance flux towers; (2) simulated CO₂ fluxes from ecosystem carbon cycle models; and (3) inverse estimates of net CO₂ fluxes inferred from atmospheric data using a global inversion of atmospheric transport.

First, the eddy covariance tower-based ecosystem carbon fluxes (Jung et al., 2009, 2011) are upscaled from direct flux observations in machine learning algorithms, and have been used for ecosystem carbon cycle studies (Beer et al., 2010; Anav et al., 2013, 2015; Bonan et al., 2011; Zscheischler et al., 2014). The data product generated spatio-temporal fields of carbon fluxes from the analysis of carbon fluxes and environmental drivers based on 224 flux tower sites (Jung et al., 2017). But the majority of eddy-covariance sites are located within the temperate climates. The sampling of boreal and arctic environments are sparse in the current

eddy-covariance network (Williams et al., 2009). Moreover, in the generation of net ecosystem exchange (NEE), the impact of disturbances such as land-use change and fires are not considered (Jung et al., 2017). This estimate of NEE between the atmosphere and terrestrial ecosystems did not fully account for the carbon balance of ecosystems.

Second, ecosystem carbon cycle model is an alternative method to estimate large-scale net ecosystem carbon fluxes (Fisher et al., 2014a; Luo et al., 2015). Despite considerable progress over the past decade, ecosystem models still do not include or realistically simulate some processes such as permafrost carbon dynamics (Koven et al., 2011; McGuire et al., 2018), and fire disturbances (Bond-Lamberty et al., 2007; Beck and Goetz, 2011) over arctic and boreal ecosystems. Furthermore, there remains substantial uncertainties in simulating northern high latitude carbon cycle (Luo et al., 2012, 2015; Huntzinger et al., 2012; Lovenduski and Bonan, 2017). Large discrepancies exist not only in the direction, but also in the spatial distribution of net surface carbon fluxes (Xia et al., 2017; McGuire et al., 2012; Fisher et al., 2014b). For example, the inter-comparison of different models showed that the simulated arctic tundra CO₂ exchange for the period 1990 to 2006 ranges from 1 TgC yr⁻¹ (almost carbon neutral) to a carbon sink of 255 TgC yr⁻¹ (McGuire et al., 2012). We have also shown that most models of terrestrial ecosystems do not correctly reproduce the response of ecosystem productivity to temperature variation over the last three decades (see Supplementary Fig. 15).

Third, inversions of atmospheric CO₂ concentration provide a diver-down approach to estimate net surface CO₂ fluxes. The current relatively complete set of inverse model results, however, diverge on net ecosystem exchange (NEE) at high latitudes. McGuire et al. (2012) compared eight inverse models and found that inverse estimates of NEE ranged from a sink of 331 TgC yr⁻¹ to a source of 173 TgC yr⁻¹ for the period 1990 to 2006. Moreover, inverse estimates of net surface CO₂ fluxes, in particular the inter-annual variability, are sensitive to the composition of the CO₂ observing network used (Rödenbeck et al., 2003; Chevallier et al., 2012; Peylin et al., 2013). This would complicate our understanding of temporal changes in inter-annual temperature sensitivity of fluxes, based on inverse systems that generally used a continuously growing number of CO₂ observing sites throughout the whole time period (Peylin et al., 2013), or included dense CO₂ sampling from satellite observations during the post-CO₂ satellite period (Sellers et al., 2018).

The actual CO₂ fluxes derived from above-mentioned approaches could suffer from large uncertainties at high latitudes, and are therefore not well suited for our analysis. The long-term atmospheric CO₂ observation station has provided complementary monitoring of large-scale terrestrial carbon cycle from the atmosphere, since the seasonal rise and decline of atmospheric CO₂ concentration is due principally to the metabolic activity of terrestrial biosphere (Keeling et al., 1996; Randerson et al., 1999; Graven et al., 2013). A growing body of studies have used the current longest-running atmospheric CO₂ stations to understand temporal changes in regional-to-global carbon cycles and their associations with climate

drivers (Wang et al., 2014; Graven et al., 2013; Forkel et al., 2016; Piao et al., 2017; Liu et al., 2018).

As the reviewer correctly pinpointed, concentrations and fluxes are two different parameters, since atmospheric CO₂ concentrations integrate net surface CO₂ fluxes from different regions through atmospheric transport. Although atmospheric CO₂ concentration is not a direct measure of terrestrial carbon fluxes, changes in CO₂ concentration at the Barrow station should be a good indicator for changes in terrestrial carbon fluxes over boreal and arctic regions. For example, previous studies identified that nearly all of the seasonality of CO₂ concentration at the Barrow site can be attributed to terrestrial carbon fluxes, with the largest contributions mainly from arctic and boreal regions (Graven et al., 2013). In addition, our transport simulations using the FLEXPART Lagrangian particle dispersion model also showed that the summer flux footprint of the Barrow station was mainly restricted to the regions of Siberia and Alaska (see Supplementary Fig. 4), implying that the CO₂ signal at Barrow station mainly reflects summer CO₂ fluxes from boreal and arctic regions.

Furthermore, we have performed an ensemble of transport simulations to demonstrate the linkage between change in summer CO₂ drawdown and that in summer terrestrial CO₂ fluxes. We used historical net biome productivity (NBP) simulations during the period 1979–2012 from terrestrial ecosystem models, which participate in the historical climatic carbon-cycle model comparison project (Trendy) and the NBP dataset from MACC, to generate an

ensemble of eight different NBP changes. We then use LMDZ atmospheric transport model to transform the simulated NEP into a point estimate of CO₂ concentration at Barrow station.

Our results showed that simulated changes in summer CO₂ drawdown (SCD) from the period 1996–2012 to 1979–1995 at Barrow station are strongly correlated with their corresponding summer NBP changes north of 50°N ($R^2 = 0.65$, $P < 0.01$) (Figure R3).

In order to respond to the reviewer's concern, we have added a section "*The linkage between atmospheric CO₂ concentration and net surface CO₂ fluxes*" into Methods of the revised MS.

“Atmospheric CO₂ concentration and net carbon fluxes are two different parameters: atmospheric CO₂ concentrations integrate net carbon fluxes from different regions through atmospheric transport. To quantify the linkage between atmospheric CO₂ concentration and net surface CO₂ fluxes, we use LMDZ atmospheric transport model to convert net CO₂ fluxes into an estimate of atmospheric CO₂ concentration at Barrow station. The transport model is nudged with winds from the European Centre for Medium-Range Weather Forecasts (ECMWF) reanalysis. We performed an ensemble of transport simulations for the period 1979–2012, using different NBP simulations from terrestrial ecosystem models participating in Trendy project and the NBP dataset from MACC inversion system. Our results showed that simulated changes in SCD from the period 1996–2012 to 1979–1995 at Barrow station are strongly correlated with their corresponding summer NBP changes north of 50°N ($R^2 = 0.67$, $P < 0.01$) (Supplementary Fig. 21). Although atmospheric CO₂ concentration was not a direct measure of terrestrial net CO₂ fluxes, change in SCD at the Barrow station is a good

indicator for change in net surface CO₂ fluxes north of 50°N .” (Line 448–462 on page 22–23)

We have also added the following sentences to stress the limitation of this study based on analysis of atmospheric CO₂ concentration at Barrow station.

“We should stress that understanding of the emerging negative temperature control on summer CO₂ uptake and its mechanisms is still limited. It would be more straightforward to use net surface CO₂ flux data, instead of atmospheric CO₂ concentration data, in depicting the relationship between carbon cycle and climate. But the availability of reliable long-term flux data is currently extremely limited by the sparsity of the in-situ observing effort over arctic and boreal regions. Continued efforts are required to increase in-situ carbon-cycle observations over these high-latitude ecosystems, to develop more mechanistic ecosystem models and to improve inverse models of assimilating CO₂ concentration, so as to provide a robust integrated estimate of net carbon exchange and its component processes (photosynthesis and respiration) over Arctic and boreal regions.” (Line 282–291 on page 14)

In addition, to be more accurate, we have changed the term “*summer CO₂ uptake*” into “*summer CO₂ drawdown*” in the revised MS. In the abstract, we have also changed “*At first glance, summer CO₂ uptake (the difference in CO₂ concentration between the first week of July and the last week of August) is significantly negatively correlated with terrestrial temperature north of 50°N interannually during 1979–2012.*” into “*At first glance, summer*

CO₂ drawdown (the difference in CO₂ concentration between the first week of July and the last week of August), used as a surrogate for summer carbon uptake, is significantly negatively correlated with terrestrial temperature north of 50°N interannually during 1979–2012.”

Figure R3. The relationship between SCD change and terrestrial net biome productivity

(NBP) change. We use LMDZ atmospheric transport model to convert each NBP map into an estimate of atmospheric CO₂ concentration at Barrow station. For each NBP, the corresponding SCD is calculated from the simulated atmospheric CO₂ concentration using LMDZ. The changes in NBP and SCD are computed as the difference between the period 1996–2012 and 1979–1995. Note that the model LPJ was recognized as an outlier and is then

not included in the correlation analysis.

[Comment 2] *L31-34. Summer uptake is not shown. Net uptake is the result of ecosystem respiration and photosynthesis. Respiration is only known from model outputs and ecosystem carbon models for the Arctic are known to vary dramatically. Averaging a number of disparate model outputs does not necessarily give the correct resultant flux.*

[Response] As the reviewer correctly pinpointed, the use of the multimodel mean seems to provide a more robust estimate of heterotrophic respiration than any single model, but the accuracy of which might be questioned since no observation information is contained.

Moreover, these models do not include or realistically simulate some processes such as permafrost carbon dynamics (Koven et al., 2011; McGuire et al., 2018), and fire disturbances (Bond-Lamberty et al., 2007; Beck and Goetz, 2011) over arctic and boreal ecosystems. In addition, our results showed that most models do not correctly reproduce the response of ecosystem productivity to temperature variability over the last three decades (see Supplementary Fig. 15 in the revised MS). Because of a tight coupling between productivity and respiration, it is then questionable to use modeled respiration to quantify changes in the response of terrestrial respiration to temperature.

Here we therefore analyzed a global respiration dataset that integrates global soil respiration database (SRDB) (Bond-Lamberty and Thomson, 2010; version 3) in a climate-driven empirical model of soil respiration (Hashimoto et al., 2015; referenced as ‘H2015’ hereafter).

We applied the partial correlation analysis on the H2015 respiration dataset and found a significant positive correlation between H2015 respiration and summer temperature in both earlier ($R_{HR-T} = 0.85$, $P < 0.01$) and later periods ($R_{HR-T} = 0.57$, $P < 0.05$) (Figure R4). This result suggests that change in the strength of the correlation would not be mainly responsible for the shift in R_{SCD-T} . However, we should inform that this dataset still suffers from uncertainties (Bond-Lamberty and Thomson, 2010; Bond-Lamberty, 2018). Our understanding of the mechanism responsible for emerging negative temperature control on summer CO₂ uptake from the respiration perspective might be limited. Rigorous testing will require systematic and repeated measurements of respiration at larger spatial scales in the future studies. This limitation has been acknowledged in conclusion part of the revised MS (see also responses to *Comment 1* by **Reviewer #2**).

Thanks to your suggestion, we mainly used SRDB dataset to explore change in the relationship between respiration and temperature in the revised MS. We removed the analyses related to respiration field based upon TRENDY models and MACC inversion system. Therefore, we have replaced “*We inferred the ecosystem respiration (RECO) field by subtracting GPP based on flux-tower data²⁸ from net biome production (NBP) derived from the net CO₂ flux from the Monitoring Atmospheric Composition and Climate (MACC) atmospheric CO₂ inversion³⁹. The interannual partial correlation of summer RECO with summer temperature (R_{RECO-T}), while controlling for the effects of precipitation and cloudiness, was significant ($P < 0.01$) for both study periods, but the coefficients of 0.78 ± 0.13 for 1979–*

1995 and 0.72 ± 0.05 for 1996–2012 (Supplementary Fig. 18a) do not suggest a correlation becoming more positive in the later period. We also calculated heterotrophic respiration (HR) by subtracting satellite-derived NPP²⁷ from the MACC inversion net CO₂ flux and then calculated the correlations R_{HR-T} . We obtained results similar to those for R_{RECO-T} (Supplementary Fig. 18b). We also used simulated HR from the ensemble of nine models of terrestrial carbon cycles that provided bottom-up estimates of carbon cycle processes for the Global Carbon Project (see Methods). R_{HR-T} was significant for both study periods for all models except CLM 4.5 (Supplementary Fig. 18b).” with “We used a global respiration dataset that integrates a global soil respiration database⁸ with a climate-driven empirical model of soil respiration⁴⁰. The interannual partial correlation of summer heterotrophic respiration (HR) with summer temperature (R_{HR-T}), while controlling for the effects of precipitation and cloudiness, was significant for both study periods. The coefficients are 0.85 ± 0.11 for 1979–1995 ($P < 0.01$) and 0.57 ± 0.16 for 1996–2012 ($P < 0.05$), respectively (Supplementary Fig. 20).” (Line 236–241 on page 12)

Figure R4. The time series of detrended anomaly of summer heterotrophic respiration (HR) derived from Hashimoto et al. (2015) (black line) and summer temperature (T, red line) calculated as the average for July and August across ecosystems north of 50°N. The inset illustrates the interannual partial correlation coefficient between HR and T. * and ** indicates that partial correlation coefficient is statistically significant at $P < 0.05$ and $P < 0.01$, respectively.

[Comment 3] L 63-68. *Summer CO₂ drawdown maybe be an “indicator” of CO₂ uptake.*

However, the authors have not made the definitive quantitative link between SCD and net CO₂ uptake. This is a critical step missing from this analysis and presentation.

[Response] We strongly agree with the reviewer that summer CO₂ drawdown (SCD) should be used as an indicator of net CO₂ uptake. We have accordingly performed an ensemble of transport simulations to demonstrate a strong correlation between change in summer CO₂

drawdown and that in summer terrestrial CO₂ fluxes north of 50°N (see responses to

Comment 1 by Reviewer #2).

We further performed LMDZ atmospheric transport model experiments to quantify the linkage between summer CO₂ drawdown (SCD) and net CO₂ uptake based on perturbations of surface fluxes. Specifically, we increased summer surface fluxes by a scaling value so that summer NBP north of 50°N increased by 1 PgC for each year of the period 1979–2012. For a given year, the scaling value for each pixel north of 50°N is calculated as the ratio of summer NBP to one unit of PgC.

To understand the uncertainty related to the choice of different NBP, we considered 7 modeled NBP from terrestrial ecosystem models participating in TRENDY project and the NBP dataset from MACC inversion system. Our results show that on average, one unit increase of PgC in terrestrial NBP north of 50°N can lead to an increase of 5.49 ± 2.70 ppm in SCD at the Barrow station (Figure R5). We should inform that averaging an ensemble of modeled results does not necessarily give the correct sensitivity. The accurate estimate of this sensitivity relies on the accuracies of estimates of net surface fluxes and atmospheric transport model.

In order to resolve the reviewer's concern, we have added a section "*The linkage between atmospheric CO₂ concentration and net surface carbon fluxes*" into Methods of the revised MS.

"We performed further transport simulations to estimate the linkage between SCD and net surface CO₂ fluxes based on perturbations of the fluxes. Specifically, for a given summer NBP map, we performed both control and perturbed simulations using LMDZ transport model. In the perturbed case, we increased summer NBP by a scaling value so that summer NBP north of 50°N increased by 1 PgC for each year of the period 1979–2012. For a given year, the scaling value for each pixel north of 50°N is calculated as the ratio of summer NBP to one unit of PgC. The sensitivity of SCD change to NBP change is then computed as the difference of SCD between control and perturbed simulations, relative to one unit increase of PgC in terrestrial NBP north of 50°N.

To understand the uncertainty related to the choice of different NBP maps, we considered different modeled NBP maps from Trendy project and the NBP from MACC inversion system. Our results show that, on average, one unit increase of PgC in terrestrial NBP north of 50°N can lead to an increase of 5.49 ± 2.70 ppm in SCD at the Barrow station (Supplementary Fig. 22). Note that averaging an ensemble of modeled results does not necessarily give the correct sensitivity. The accurate estimate of this sensitivity relies on the accuracies of estimates of net surface CO₂ fluxes and atmospheric transport model." (Line 464–480 on page 23–24)

Figure R5. The model estimate of sensitivity of SCD change to NBP change based on perturbations of surface fluxes. For each NBP map, we performed control and perturbed simulations. In the perturbed simulation, we increased summer surface fluxes by a scaling value so that summer NBP north of 50°N increased by one unit of PgC for each year of the period 1979–2012. We use LMDZ atmospheric transport model to convert each NBP map into an estimate of atmospheric CO₂ concentration at Barrow station. For each NBP map, the SCDs from control and perturbed simulations are obtained, respectively. The sensitivity is then computed as their difference relative to one unit increase of PgC in terrestrial NBP north of 50°N.

[Comment 4] L 126-128. *Randomly selecting 14 of 16 years of data seems pointless since almost all of the data is being used in each analysis. The full data sets should be used.*

[Response] As the reviewer mentioned, the full dataset was used in the main analysis except in the section “*Robustness tests*”. In order to understand whether the probability of detecting a

shift in R_{SCD-T} over the last three decades is fortuitous or not, we therefore randomly select 14 years from 16 years in both earlier (1979–1995) and later periods (1996–2012). To clarify this, we have changed “*We analyzed the frequency distribution of R_{SCD-T} obtained by randomly selecting 14 years for each period and found that R_{SCD-T} decreased from -0.42 ± 0.09 during 1979–1995 to -0.64 ± 0.10 during 1996–2012 (Fig. 1a)*” to “*To test whether the observed shift in R_{SCD-T} is real or an artifact caused by extreme years, we analyzed the frequency distribution of R_{SCD-T} obtained by randomly selecting 14 years for each period and found that R_{SCD-T} decreased from -0.42 ± 0.09 during 1979–1995 to -0.64 ± 0.10 during 1996–2012 (Fig. 1a)*”.

(Line 128–131 on page 7)

[Comment 5] L 174. *As stated in L 180, NDVI is only a proxy for productivity. They are not synonymous.*

[Response] We have changed “*so our first hypothesis is that summer plant productivity has become less positively responsive to temperature*” into “*so our first hypothesis is that summer vegetation activities has become less positively responsive to temperature*”. (Line 174–175 on page 9)

[Comment 6] L 186–189. *These are both indirect methods. The satellite derived NPP model has large uncertainties. The GPP data is poorly constrained by data.*

[Response] We agree with the reviewer that there are uncertainties in satellite-derived net primary productivity (NPP) and the upscaled gross primary productivity (GPP) data from a global network of eddy-covariance flux towers.

To further support our conclusion, we also used solar-induced chlorophyll fluorescence (SIF). SIF reflects an integrative photosynthetic signal from the molecular origin, and is increasingly used as a physiological-based proxy for GPP (Daumard et al., 2010; Frankenberg et al., 2011; Sun et al., 2017; Li et al., 2018). For example, Sun et al. (2017) used a high-resolution SIF dataset to show that there is almost a linear relationship between SIF and gross primary productivity at eddy-flux site in diverse biomes, further supporting the utility of SIF in revealing photosynthetic activities.

Here we used an extended satellite-derived SIF data during the period 2001–2012 (Zhang et al., 2018). This dataset is generated from a trained neural network that integrates surface reflectance from the MODerate-resolution Imaging Spectroradiometer (MODIS) and SIF from Orbiting Carbon Observatory-2 (OCO-2). The extended SIF dataset available for the all-sky condition has a spatial and temporal resolution of 0.05 degree and 4 days, respectively. Since this dataset is only available after 2001, we then only analyzed the correlation of SIF with temperature across ecosystems north of 50°N during the period 2001–2012. Our analysis suggested that their correlation was not significant during the period 2001–2012 ($R_{SIF-T} = -0.19$, $P = 0.61$) (Figure R6), tentatively supporting our main conclusion.

In the revised MS, we have also added the analysis of SIF.

“In addition, analysis of an extended solar-induced chlorophyll fluorescence (SIF) dataset³⁶ also showed that the correlation between SIF and temperature across ecosystems north of 50°N was not significant during the period 2001–2012 ($R_{SIF-T} = -0.19$, $P = 0.61$) (Supplementary Fig. 14).” (Line 187–190 on page 9–10)

We have added the following text related to SIF into “Methods” section.

“Solar-induced chlorophyll fluorescence (SIF) that reflects photosynthetic signals from the molecular origin is increasingly used as a physiological-based proxy for gross primary productivity (GPP)⁵³⁻⁵⁶. Here we used an extended satellite-derived SIF data set for the period 2001–2012³⁶. This data set is generated by a trained neural network that integrates surface reflectance from the MODerate-resolution Imaging Spectroradiometer (MODIS) and SIF from Orbiting Carbon Observatory-2 (OCO-2). The extended SIF data set, which is available for the all-sky condition, has a spatial resolution of 0.05 degrees and a temporal resolution of 4 days.” (Line 402–409 on page 20–21)

In addition, since satellite-based NPP could suffer from certain uncertainties, we have then removed the Figure 3 (NPP is presented in this figure) in the revised MS.

Figure R6. The partial correlation between solar-induced chlorophyll fluorescence (SIF) and summer temperature for the period 2001–2012. Here we randomly selected 9 years from the time series for partial correlation analysis and the histogram shows the frequency distribution of the partial correlation coefficient between SIF and summer temperature. The magenta line illustrates the significance level at $P < 0.05$.

[Comment 7] L193-200. *The number of hours when canopy photosynthesis is reduced by high canopy temperature has not been shown. Even on warm days, much of the day is below the temperature optimum for photosynthesis.*

[Response] The actual high temperature stress that plants experience needs the definition of photosynthetic optimum temperature. However, most of our knowledge about the optimum temperature stems from site-level measurements (Niu et al., 2008; Way and Yamori, 2014).

We still lack essential knowledge about optimum temperatures at broad spatial scale.

Therefore, in this study, we defined extreme warm events that are potentially unfavorable for photosynthesis through counting the number of days or hours with daily or hourly mean

temperature exceeding a certain threshold. We should inform that this approach is just used as an approximation to indicate the potential risk of high temperature stress on photosynthesis.

Following your suggestion, we have also calculated changes in number of extreme warm hours. For each day of the summer season (July and August), we extracted the maximum temperature and its two neighbors based on 3-hourly temperature data from ERA-Interim data (http://www.eu-watch.org/gfx_content/documents/README-WFDEI.pdf). These temperatures occurring throughout the entire study period 1979–2012 were sorted in ascending to determine the threshold value for the 90th percentile. The number of hours with temperatures exceeding 90th percentile were then referred to as extreme warm hours. The main conclusion is robust to the use of sub-daily temperature records in defining extreme warm temperatures. For example, the number of extreme warm hours increased during the later period (Figure R7a). The patterns of the changes of R_{NDVI-T} were roughly consistent with those of the number of extreme warm hours, particularly Alaska and eastern Siberia constituting the main footprint area of summer CO₂ changes at Barrow (Supplementary Fig. 4). NDVI in these areas had a significant negative partial correlation with the number of extreme warm hours when the data were statistically controlled for the effect of mean summer temperature (Figure R7b).

In order to resolve the reviewer's concern, we have added the following sentence into the revised MS and Figure R7 into the Supplementary text.

“The results are robust to the use of sub-daily temperature records in the definition of extreme warm temperatures (Supplementary Fig. 17).” (Line 206–207 on page 10)

Figure R7. Spatial distribution of the changes in extreme warm hours during July and

August (a) and the linkage between summer NDVI and extreme warm hours (b). For

each day of the summer season (July and August), we extracted the maximum temperature

and its two neighbors based on 3-hourly temperature data from ERA-Interim data. The

3-hourly temperatures occurring throughout the entire study period 1979–2012 were sorted in

ascending to determine the threshold value for the 90th percentile. The number of hours with

temperatures exceeding 90th percentile were then referred to as extreme warm hours (TX90p).

The changes are the difference between the 1996–2012 and 1979–1995 periods. The R_{NDVI-}

$-r_{TX90p}$ is defined as the partial correlation coefficient of NDVI against TX90p, whilst controlling for the effect of summer temperature. The gray dots indicate that the partial correlation coefficient is statistically significant ($P < 0.05$).

[Comment 8] *Water stress might contribute to reduced photosynthesis. This is speculative as not data is shown. L 209-212. Soil moisture and drought are two separate things. High temperature can cause drought even with high soil moisture. This is likely to occur, e.g., when root resistance is high due to low soil temperatures as occurs in Arctic soils.*

[Response] As the reviewer stated, water stress could adversely affect rates of photosynthesis due to restricted CO₂ diffusion into the leaf resulted from stomatal closure, and inhibition of photosynthetic metabolism (Lawlor and Cornic, 2002; Flexas et al., 2006; Chaves et al., 2009). Plant water stress is a combined function of soil water supply and atmospheric demand for water. Our analysis of soil moisture changes showed that summer soil moisture were slightly higher during the later than the earlier period (see Supplementary Fig. 18), suggesting that soil water supply is unlikely to cause water stress to plant growth.

Increased temperature could induce or exacerbate plant water stress through increasing atmospheric deficits to the degree which plants either lose water at a faster rate or close stomata. Here we also analyzed changes in the atmospheric vapour pressure deficit (VPD), which is indicative of atmospheric demand for water, between the earlier and later period. Our results showed that atmospheric demand for water generally increased over the main footprint

area of summer CO₂ changes at Barrow (Figure R8a). But NDVI in these areas had either a positive or non-significant negative partial correlation with VPD changes when the data were statistically controlled for the effect of mean summer temperature (Figure R8b). It suggested that increased atmospheric demand for water over high-latitude ecosystems has a relatively low probability of imposing water stress constraints on plant growth. Therefore, changes in VPD might not account for the decrease in R_{NDVI-T} .

As the reviewer mentioned, soil moisture and drought are two separate things. We have therefore changed “*These values of soil moisture in summer were slightly higher during the later than the earlier period (Supplementary Fig. 17), indicating no increase in summer drought. Thus, changes in soil moisture alone did not account for the decrease in R_{NDVI-T} .*” into “*These values of soil moisture in summer were slightly higher during the later than the earlier period (Supplementary Fig. 18), suggesting that changes in soil moisture alone did not account for the decrease in R_{NDVI-T} .*” (Line 216–219 on page 11)

In addition, we also added the analysis of VPD change in the revised MS.

“In addition, increased temperature could induce plant water stress, through increasing atmospheric deficits, to such a degree that plants either lose water at a faster rate or close stomata. We also analyzed changes in the atmospheric vapour pressure deficit (VPD), which is indicative of atmospheric demand for water, between the earlier and later period. Our results showed that atmospheric demand for water generally increased over the main

footprint area of summer CO_2 changes at Barrow (Supplementary Fig. 19a). But NDVI in these areas had either a positive or non-significant negative partial correlation with VPD changes when the data were statistically controlled for the effect of mean summer temperature (Supplementary Fig. 19b). This result suggests that increased atmospheric demand for water has a relatively low probability of imposing water stress constraints on plant growth over high latitude ecosystems. Therefore, change in VPD should not be the main cause of the decrease in R_{NDVI-T} .” (Line 219–230 on page 11)

Figure R8. Spatial distribution of changes in vapor pressure deficit (VPD) during July and August (a) and the linkage between summer NDVI and VPD (b). The VPD changes are the difference between the 1996–2012 and 1982–1995 periods. The $R_{NDVI-VPD}$ is defined as

the partial correlation coefficient of NDVI against VPD, whilst controlling for the effect of summer temperature. The gray dots indicate that the partial correlation coefficient is statistically significance ($P < 0.05$).

[Comment 9] L228-229. *These models perform poorly against data. The mean or ensemble average of a number of disparate model outputs, one does not necessarily yield a true estimate of HR.*

[Response] Thanks to your suggestion, we mainly used SRDB dataset to explore change in the relationship between respiration and temperature in the revised MS. We removed the analyses related to modeled respiration (see details in response to **Comment 2** by *Reviewer #2*)

[Comment 10] L231-234. *If respiration continues to be depending on temperature, and if temperature continues to rise, it is unclear why this might not cause the purported decrease in summer NEE.*

[Response] As the reviewer proposed, the finding that terrestrial respiration continues to be depending on temperature suggested an increase of temperature during the later period would continue to stimulate terrestrial respiration and thus decrease net carbon uptake. In this study, we are mainly concerned with change in inter-annual correlation between (indicator of) summer net carbon uptake and temperature in the last 3 decades, instead of change in (indicator of) summer carbon uptake itself. To explain the weakened temperature dependence

of (indicator of) summer carbon uptake, we proposed and tested the following two hypotheses:
a weakened temperature dependence of (indicator of) plant productivity and a loss of
inter-annual correlation between terrestrial respiration and temperature.

Thanks to your suggestion, we have changed “*We tested the hypothesis that the increased response of respiration to temperature was responsible for the change in R_{SCD-T} .*” into “*We tested the hypothesis that the loss of temperature dependence of terrestrial respiration was responsible for the change in R_{SCD-T} .*” (Line 234–236 on page 12).

In addition, we have also changed “*These analyses suggested that respiration continued to respond significantly positively to temperature in both study periods, but that the response was not stronger during the later period, implying that the change in R_{SCD-T} was not due to changes in the response of terrestrial respiration to temperature.*” to “*Our analysis suggested that respiration continued to respond significantly positively to temperature in both study periods, albeit that the response was not stronger during the later period, implying that the change in R_{SCD-T} was not due to changes in the response of terrestrial respiration to temperature.*” (Line 242–245 on page 12)

[Comment 11] L236-238. Discussion. The statement that “*We demonstrated that the effect of temperature on summer CO_2 uptake in arctic and boreal ecosystems has been altered in the last three decades*”. There is no presentation of the actual rates of CO_2 uptake over the last

three decades. There is inference, but not verified data. Therefore, this statement cannot be made. Only statements about the change in rates of drawdown with summer temperature can be made. The paper overstates the conclusions possible.

[Response] Thanks to **Comment 1**, we performed an ensemble of transport simulations to show that there is a strong relationship between change in summer CO₂ drawdown and change in summer net surface carbon fluxes north of 50°N (see detailed in responses to **Comment 1** by **Reviewer #2**). Then it is relatively safe to imply changes in summer terrestrial carbon fluxes over arctic and boreal ecosystems from changes in summer CO₂ drawdown at the Barrow station.

Following your suggestion, we have changed the sentence “*We demonstrated that the effect of temperature on summer CO₂ uptake in arctic and boreal ecosystems has been altered in the last three decades.*” to “*We demonstrated that the effect of temperature on summer CO₂ drawdown has changed in the last three decades, implying a shift in the responses of summer CO₂ uptake to warming in Arctic and boreal ecosystems.*”. (Line 248–250 on page 12)

[Comment 12] L257. *The statement that “High-latitude ecosystems have not shifted from a carbon sink to a carbon source” cannot be made base on this paper or on the one paper cited that uses inference. There is considerable literature and uncertainty in this area, and it is insufficient to cite a single paper, especially one that is not definitive.*

[Response] We strongly agree with the reviewer's comment. In order not to mislead the readers, we have changed the sentences "*High-latitude ecosystems have not shifted from a carbon sink to a carbon source¹⁹, suggesting that the large temperature-induced shift in net CO₂ uptake during the main period of carbon uptake did not outweigh the increase in carbon sinks due e.g. to the gradual increasing atmospheric concentration of CO₂⁴², or increased nitrogen deposition⁴³.*" to "*Our finding will add further complexity to the estimation of carbon balance over high-latitude ecosystems in a warming world. The large temperature-induced shift in carbon uptake during the main period of carbon uptake would reduce positive effects of the gradual increasing atmospheric concentration of CO₂⁴³, and increased nitrogen deposition⁴⁴ on net CO₂ uptake.*" (Line 268–272 on page 13)

Reviewer #3

[General Comment] *The revised manuscript addresses the concerns regarding transport from lower latitudes which I had with the original through some additional simulations using longer integrations of the Lagrangian parcel model. In addition, the authors have added some important context in the recent literature which were missing in the original manuscript.*

In addition to my own concerns, the authors have done a nice job in addressing the concerns of the other reviewer, particularly with regard to aliasing of phase and amplitude effects in the time series.

I find that the revised manuscript is an important and timely contribution to the emerging understanding of carbon-climate feedbacks in high-latitude ecosystems, and recommend that it be published.

[Response] Thanks for your great interests on our work. Previous comments have significantly improved our MS.

References

- Anav, A. *et al.* Evaluating the land and ocean components of global carbon cycle in the CMIP5 Earth System Models. *J. Climate*. **26**, 6801–6843 (2013).
- Anav, A. *et al.* Spatiotemporal patterns of terrestrial gross primary production: A review. *Rev. Geophys.* **53**, 785–818 (2015).
- Beer, C. *et al.* Terrestrial gross carbon dioxide uptake: global distribution and covariation with climate. *Science* **329**(5993), 834-838 (2010)
- Beck, P. S. A. and Goetz, S. J. Satellite observations of high northern latitude vegetation productivity changes between 1982 and 2008: ecological variability and regional differences. *Environ. Res. Lett.* **6**, 045501 (2011)
- Bonan, G. B. *et al.* Improving canopy processes in the Community Land Model version 4 (CLM4) using global flux fields empirically inferred from FLUXNET data. *J. Geophys. Res.-Biogeo.* **116** (G2), G02014 (2011)
- Bond-Lamberty, B. *et al.* Fire as the dominant driver of central Canadian boreal forest carbon balance. *Nature* **450**, 89–93 (2007)
- Bond-Lamberty, B. and Thomson, A. Temperature-associated increases in the global soil respiration record. *Nature* **464**(7288), 579-582 (2010)
- Bond-Lamberty, B. Data sharing and scientific impact in eddy covariance research. *J. Geophys. Res.-Biogeo* **123**(4) (2018)
- Chaves, M. M. *et al.* Photosynthesis under drought and salt stress: regulatory mechanisms from whole plant to cell. *Ann. Bot-London* **103**(4), 551-560 (2009)

- Chevallier, F., *et al.* What eddy-covariance flux measurements tell us about prior errors in CO₂-flux inversion schemes. *Global Biogeochem. Cycles*, **26**, GB1021 (2012)
- Daumard, F. *et al.* A field platform for continuous measurement of canopy fluorescence. *IEEE T. Geosci. Remote* **48**, 3358-3368 (2010).
- Flexas, J. *et al.* Decreased Rubisco activity during water stress is not induced by decreased relative water content but related to conditions of low stomatal conductance and chloroplast CO₂ concentration. *New Phytol.* **172**(1), 73-82 (2006)
- Fisher, J. B. *et al.* Modeling the terrestrial biosphere. *Annu. Rev. Env. Resour.* **39**, 91-123 (2014a)
- Fisher, J. B. *et al.* Carbon cycle uncertainty in the Alaskan Arctic. *Biogeosciences* **11**, 4271-4288 (2014b)
- Forkel, M. *et al.* Enhanced seasonal CO₂ exchange caused by amplified plant productivity in northern ecosystems. *Science* **6274**, 696-699 (2016)
- Frankenberg, C. *et al.* New global observations of the terrestrial carbon cycle from GOSAT: Patterns of plant fluorescence with gross primary productivity. *Geophys. Res. Lett.* **38**(17), L17706 (2011)
- Graven, H. D. *et al.* Enhanced Seasonal Exchange of CO₂ by Northern Ecosystems since 1960. *Science* **341**, 1085–1089 (2013).
- Hashimoto, S. *et al.* Global spatialtemporal distribution of soil respiration modeled using a global database. *Biogeosciences* **12**(13), 4121-4132 (2015)
- Huntzinger, D. N. *et al.* North American Carbon Project (NACP) Regional Interim Synthesis:

- Terrestrial Biospheric Model Intercomparison, *Ecol. Model.* **224**, 144–157 (2012).
- Jung, M. *et al.* Towards global empirical upscaling of FLUXNET eddy covariance observations: validation of a model tree ensemble approach using a biosphere model. *Biogeosciences* **6**, 2001-2013 (2009)
- Jung, M. *et al.* Global patterns of land-atmosphere fluxes of carbon dioxide, latent heat, and sensible heat derived from eddy covariance, satellite, and meteorological observations. *J. Geophys. Res.* **116**, G00J07 (2011).
- Jung, M. *et al.* Compensatory water effects link yearly global land CO₂ sink changes to temperature. *Nature* **541**, 516-520 (2017)
- Keeling, C. D. *et al.* Increased activity of northern vegetation inferred from atmospheric CO₂ measurements. *Nature* **382**, 146-149 (1996)
- Koven, C. D. *et al.* Permafrost carbon-climate feedbacks accelerate global warming. *P. Natl. Acad. Sci. USA* **108**(36), 14769-14774 (2011)
- Lawlor, D. W. and Cornic G. Photosynthetic carbon assimilation and associated metabolism in relation to water deficits in higher plants. *Plant Cell Environ.* **25**(2) (2002)
- Luo Y. Q. *et al.* A framework for benchmarking land models. *Biogeosciences* **9**, 3857–3874 (2012)
- Luo, Y. Q. *et al.* Predictability of the terrestrial carbon cycle. *Glob. Change Biol.* **21**, 1737-1751 (2015)
- Lovenduski, N. S. and Bonan, G. B. Reducing uncertainty in projections of terrestrial carbon uptake. *Environ. Res. Lett.* **12**, 101001 (2017)

- Li, X. *et al.* Solar-induced chlorophyll fluorescence is strongly correlated with terrestrial photosynthesis for a wide variety of biomes: First global analysis based on OCO-2 and flux tower observations. *Glob. Change Biol.* **24**(9) 3990-4008 (2018)
- Liu, D. *et al.* Decelerating autumn CO₂ release with warming induced by attenuated temperature dependence of respiration in northern ecosystems. *Geophys. Res. Lett.* **45**(11) (2018)
- McGuire, A. D. *et al.* An assessment of the carbon balance of Arctic tundra: comparisons among observations, process models, and atmospheric inversions. *Biogeoscience* **9**, 3185-3204 (2012)
- McGuire, A. D. *et al.* Dependence of the evolution of carbon dynamics in the northern permafrost region on the trajectory of climate change. *P. Natl. Acad. Sci. USA* doi: <https://doi.org/10.1073/pnas.1719903115> (2018)
- Niu, S. *et al.* Thermal optimality of net ecosystem exchange of carbon dioxide and underlying mechanisms. *New Phytol.* **194**(3), 775-783. (2012)
- Piao, S. L. *et al.* Weakening temperature control on the interannual variations of spring carbon uptake across northern lands. *Nature Clim. Change* **7**, 359-363 (2017)
- Peylin, P. *et al.* Global atmospheric carbon budget: results from an ensemble of atmospheric CO₂ inversions. *Biogeosciences* **10**, 6699-6720 (2013)
- Rödenbeck, C. *et al.* CO₂ flux history 1982-2001 inferred from atmospheric data using a global inversion of atmospheric transport. *Atmospheric Chemistry and Physics*, European Geosciences Union, **3** (6), 1919-1964 (2003).

- Randerson, J. T. *et al.* Increases in early season ecosystem uptake explain recent changes in the seasonal cycle of atmospheric CO₂ at high northern latitudes. *Geophys. Res. Lett.* **26**, 2765–2768 (1999).
- Sun, Y. *et al.* OCO-2 advances photosynthesis observation from space via solar-induced chlorophyll fluorescence. *Science* **358** (6360), eaam5747 (2017)
- Sellers, P. J. *et al.* Observing carbon cycle-climate feedbacks from space. *P. Natl. Acad. Sci. USA* doi: <https://doi.org/10.1073/pnas.1716613115> (2018)
- Wang, X. H. *et al.* A two-fold increase of carbon cycle sensitivity to tropical temperature variations. *Nature* **506**, 212-215. (2014)
- Way, D. A. and Yamori, W. Thermal acclimation of photosynthesis: on the importance of adjusting our definitions and accounting for thermal acclimation of respiration. *Photosynth. Res.* **119**(1-2) 89-100 (2014)
- Williams, M. *et al.* Improving land surface models with FLUXNET data. *Biogeosciences* **6**, 1341-1359 (2009)
- Xia, J. Y. *et al.* Terrestrial ecosystem model performance in simulating productivity and its vulnerability to climate change in the northern permafrost region. *J. Geophys. Res.-Biogeophys.* **122**(2) (2017)
- Zscheischler, J. *et al.* A few extreme events dominate global interannual variability in gross primary production. *Environ. Res. Lett.* **9**, 035001 (2014)
- Zhang, Y. *et al.* A global spatially Continuous Solar Induced Fluorescence (CSIF) dataset using neural networks. *Biogeosciences Discuss.*,

<https://doi.org/10.5194/bg-2018-255> (2018)

REVIEWERS' COMMENTS:

Reviewer #1 (Remarks to the Author):

The authors have done considerable follow up analysis with regards to changes in the seasonal window of their analysis and how changes with respect to sea ice may affect seasonal C uptake over time. In particular their analysis of the diurnal temperature range is interesting in that it seems to show an inflection point and may help diagnose the mechanisms causing their emergent negative relationship between temperature and summer uptake. It would probably be more interesting to look to see how the seasonal temperature range has changed to see if there has been a greater response in respiration or photosynthesis.

Reviewer #2 is correct in their assessment that atmospheric concentration is not the same as surface flux; however, changes in the concentration of atmospheric CO₂ do provide a means of assessing changes in uptake. Unfortunately as Wang et al. point out, there is no perfect observation or modeling datasets to assess spatially explicit surface fluxes. Ultimately, they use DGVM simulations transported with an atmospheric transport model as a reasonable approach. While even these models have their faults in dealing with complex hydrology and permafrost at high latitudes, their estimates of NBP seem to at least correspond to seasonal C uptake (Fig s21). I wonder if these models show any differences in the trends of photosynthesis or respiration over time in response to changes in growing season temperature?

Wang et al. have done a thorough analysis given the sparse observations at high latitudes and I think that this analysis will inspire future work.

Reviewer #1

[Comment 1] *The authors have done considerable follow up analysis with regards to changes in the seasonal window of their analysis and how changes with respect to sea ice may affect seasonal C uptake over time. In particular their analysis of the diurnal temperature range is interesting in that it seems to show an inflection point and may help diagnose the mechanisms causing their emergent negative relationship between temperature and summer uptake. It would probably be more interesting to look to see how the seasonal temperature range has changed to see if there has been a greater response in respiration or photosynthesis.*

[Response] Following your suggestion, we analyzed changes in summer temperature range based on daily temperature from ERA-WFDEI product, which is generated by using WATCH Forcing Data methodology applied to ERA-Interim reanalysis data (Weedon et al., 2014). The summer temperature range did not show a significant trend from 1982 to 2012 when the summer was defined as July to August or as the interannually varying period from spring zero-crossing date to the day when CO₂ reaches its annual minimum (Figure R1). This analysis tentatively suggests that change in seasonal temperature range may not explain the shift in the correlation between summer carbon uptake and temperature over the past three decades.

Figure R1. Changes in summer temperature range from 1979 to 2012. The summer temperature range is calculated as the difference of maximum and minimum daily temperature for July and August (**a**) and during the period from spring zero-crossing date to the day when CO₂ reaches its annual minimum (**b**) for each year.

[Comment 2] *Reviewer #2 is correct in their assessment that atmospheric concentration is not the same as surface flux; however, changes in the concentration of atmospheric CO₂ do provide a means of assessing changes in uptake.*

Unfortunately as Wang et al. point out, there is no perfect observation or modeling datasets to assess spatially explicit surface fluxes. Ultimately, they use DGVM simulations transported with an atmospheric transport model as a reasonable approach. While even these models have their faults in dealing with complex

hydrology and permafrost at high latitudes, their estimates of NBP seem to at least correspond to seasonal C uptake (Fig s21). I wonder if these models show any differences in the trends of photosynthesis or respiration over time in response to changes in growing season temperature?

[Response] Following your suggestion, for each model, we calculated partial-correlation coefficients between net primary productivity (NPP), heterotrophic respiration (HR) and net ecosystem productivity (NEP) against summer temperature whilst controlling for the effects of summer precipitation and cloud cover during the earlier (1979–1995) and later (1996–2012) periods (Figure R2). Only two models captured the loss of the positive correlation between summer NPP and summer temperature (Figure R2a). Analysis of modeled HR suggested that respiration continued to respond significantly positively to temperature for most of models in both study periods (Figure R2b). This model result is consistent with our analysis based upon a global soil respiration database. By contrast, only three models captured the emerging negative response of summer NEP to temperature in the later period (Figure R2c). These results highlighted that the models will require better parameterizations of the processes driving the response of Arctic and boreal ecosystems to warming.

Figure R2. Inter-annual partial correlation of net primary productivity (R_{NPP-T} , a), heterotrophic respiration (R_{HR-T} , b) and net ecosystem productivity (R_{NEP-T} , c) with summer temperature for 1979–1995 (blue) and 1996–2012 (red) periods for Trendy models. The standard-deviation error bars were generated by randomly selecting 14 years for each period. °, * and ** indicates that partial correlation coefficient is statistically significant at $P < 0.1$, $P < 0.05$ and $P < 0.01$, respectively.

[Comment 3] *Wang et al. have done a thorough analysis given the sparse observations at high latitudes and I think that this analysis will inspire future work.*

[Response] Thanks a lot for your encouraging comment!